# Detecting the resilience of soil moisture dynamics to drought periods as function of soil type and climatic region

- 3 Nedal Aqel<sup>1</sup>, Jannis Groh<sup>2,3</sup>, Lutz Weihermüller<sup>2</sup>, Ralf Gründling<sup>4</sup>, Andrea Carminati<sup>1</sup>, Peter
- 4 Lehmann<sup>1</sup>
- 5 Physics of Soils and Terrestrial Ecosystems, ETH Zurich, Zurich, Switzerland
- 6 <sup>2</sup>Institute of Bio- and Geoscience IBG-3: Agrosphere, Forschungszentrum Jülich GmbH, Jülich, Germany
- 7 Biogeochemistry and Gas Fluxes, Leibniz Institute for Agricultural and Landscape Research (ZALF),
- 8 Müncheberg, Germany
- 9 <sup>4</sup>Department of Soil System Science, Helmholtz-Zentrum für Umweltforschung GmbH UFZ, Halle, Germany
- \*Corresponding author, nedal.aqel@usys.ethz.ch, Universitätstrasse 16, 8092 Zurich, Switzerland

#### 11 Abstract


















Abrupt changes in climatic conditions and land management can cause permanent shifts in soil hydraulic response to climatic inputs, impacting soil functions and established soil-climate interactions. To quantify the resilience of soil water content dynamics after abrupt changes in environmental conditions, we present a model framework combining a neural network with seasonal trend analysis (STL). Using data from a series of lysimeters from the TERrestrial ENvironmental Observatories (TERENO) - SOILCan lysimeter network, we identified changes in soil water content responses after an extremely hot and dry summer in Germany in 2018. The model incorporates meteorological variables decomposed into seasonal and long-term components along with a categorical indicator of current moisture conditions. It is trained on data from a reference site with stable soil water content response and applied to lysimeters from multiple origins exposed to contrasting climates. By analysing annual residual patterns—particularly mean bias over time—soil water content state dynamics is classified as 'stable', 'resilient', or 'changed', reflecting whether the system maintains, recovers, or diverges from its original state. We found that soils preserve the response function to environmental forcing under typical conditions but exhibit structural change when relocated to new environments, even when soil texture remains constant. The proposed method offers a scalable and non-invasive tool for tracking changes in the response of soil water content to climatic change and provides early indicators of changes in essential soil functions and soil health status.


31 32









41 42














### 1. Introduction

Soil water content plays a fundamental role in hydrological processes and land-atmosphere interactions, governing the exchange of water and energy at the Earth's surface (Seneviratne et al., 2010; Sun et al., 2025). It regulates key hydrological functions, including infiltration, runoff generation, evapotranspiration, and groundwater recharge. Through these processes, soil water content influences water availability, ecosystem productivity, and climatic conditions across local to global scales (Bogena et al., 2015; Fatichi et al., 2020). Soil water content status and related soil environmental conditions change with short- and long-term atmospheric processes. This response of soil water content to atmospheric conditions, which we define as 'soil water content response function', determines, for example, if anaerobic conditions are inhibited after heavy rainfall (by fast percolation to deeper soil layers) and if enough water remains available after dry periods for plant growth, temperature regulation, and chemical reactions. In short, the soil water content response function is a dominant factor of the soil health status. This response function is shaped by soil formation processes and reflects adaptation to the dominant climatic conditions (Kuzyakov and Zamanian, 2019; Sainju et al., 2022; Al-Shammary et al., 2025). Accordingly, a change in the response function after extreme climatic events is likely to imply a change in soil health. In the core of this study is the question how changes in the soil water content response function, and thus in soil properties and health, can be detected. The standard approach to determine the response function is to apply physically based models, for example by inverse modelling of soil water content dynamics under varying boundary conditions (Simunek et al., 2016). These models require detailed knowledge of soil hydraulic properties and extensive calibration, limiting their generalization beyond the spatial scale and the local conditions used in the calibration process (Or, 2020; Lehmann et al., 2020; O. & Orth, 2021). Additionally, such approaches are typically destructive, limiting the options of conducting a time series analysis. Moreover, these models typically assume static soil characteristics, failing to adequately represent structural changes in soil properties — such as compaction, degradation, or organic matter loss — that can substantially alter hydraulic behaviour over time (Fatichi et al., 2020; Melsen & Guse, 2021; Wankmüller et al., 2024). Most of the models also neglect or oversimplify the

hysteretic nature of the soil water retention curve, as well as seasonal changes in soil hydraulic 57 properties that can substantially alter infiltration, drainage, and plant water availability (Aqel et al., 58 2024; Hannes et al., 2016; Herbrich & Gerke, 2017). Therefore, we used a non-invasive approach based 59 on neural networks as discussed below. 60 In recent years, artificial intelligence (AI), particularly neural networks, has emerged as a promising 61 alternative for modelling complex hydrological processes (Reichstein et al., 2019). These data-driven 62 models have demonstrated the capacity to learn nonlinear relationships directly from observational 63 datasets without relying heavily on explicit physical equations (Kratzert et al., 2019; Mosavi et al., 64 2018; Shen et al., 2018). Within soil hydrology, neural networks have been used to characterise 65 hysteretic soil-water behaviour from training data, improving the representation of wetting-drying 66 cycles without explicit hysteresis parameterisation (Aqel et al., 2024). They have also been applied to soil-moisture time-series modelling using Long Short-Term Memory (LSTM) networks with recurrent 67 architectures suited to capture long-range temporal dependencies (Liu et al., 2023; O. & Orth, 2021). 68 Across diverse hydro-climatic regimes, LSTM have been shown to effectively learn nonlinear 69 70 relationships between climatic inputs and soil water content, often matching or surpassing traditional 71 physically based approaches and demonstrating strong generalisation (Kratzert et al., 2019; J. Liu et al., 72 2022, 2023; O. & Orth, 2021). 73 Independent of the chosen modelling approach, these models ignore that the soil water content response 74 function (i.e., the variations in soil water content following changes in atmospheric conditions) can 75 change at larger time scale. Experimental studies have shown that extreme events such as drought can 76 induce persistent shifts in soil water content dynamics, potentially leading to alternative stable states 77 (Robinson et al., 2016). Moreover, changes in land use — such as forest conversion to agriculture or 78 bare land — alter soil hydraulic properties, with effects on infiltration, water retention, and saturated 79 hydraulic conductivity (Fu et al., 2021; Robinson et al., 2022). Considering that texture remains 80 constant at time scales of decades, the changes in the response function are likely to be related to 81 changes in soil structure primarily.

https://doi.org/10.5194/egusphere-2025-5141 Preprint. Discussion started: 24 October 2025 © Author(s) 2025. CC BY 4.0 License.

Soil systems are subject to temporal change, yet models are often trained on historical datasets without evaluating whether the system dynamics remain stable throughout the training period (Montanari et al., 2013; Vaze et al., 2010). As a result, models may be applied to prediction settings where underlying soil—climate interactions differ from those on which the model was trained. For example, a recent study comparing different crop models using the TERENO-SOILCan set-up showed that predicting agronomic and environmental variables under different climatic conditions to those represented in the training datasets resulted in significant discrepancies between simulations and observations (Groh et al., 2022). This underscores the critical need to assess whether site-specific representations of soil—water behaviour remain valid over time (Hrachowitz et al., 2013) and highlight the need for more adaptable modelling approaches under evolving environmental conditions (Blöschl et al., 2019; Milly et al., 2008). Considering these limitations, none of the discussed approaches can capture the change in the soil water content response function. A recent modelling framework by Jarvis et al. (2024) takes a major conceptual step forward by explicitly representing soil-structure dynamics and their feedback on hydraulic behaviour. However, as the authors emphasize, such process-based models still depend on detailed observational data to constrain temporal changes in structure and hydraulic properties.

To address this gap, the present study introduces a framework based on neural networks and seasonal trend decomposition. Specifically, we quantify the change in the response function after the 2018 drought in summer, which was a Europe-wide event, but in Germany was particularly characterized by an extreme combination of high temperatures and low precipitation (Xoplaki et al., 2025). The response of soil water content on this drought will be analysed for a set of lysimeters. As shown in Fig. 1 for one of the study sites, the monthly water deficit (potential evapotranspiration minus precipitation) peaked in summer 2018 indicating the strong drought in this period.

Figure 1 Variations in climatic conditions at Selhausen (SE) expressed as difference between potential evapotranspiration (PET) and precipitation (P) cumulated over the precedent 30 days (one month). The extreme summer 2018 is manifested by a maximum monthly deficit of ~150 mm. Details on the calculation of PET are provided in section 2.1.1.

The general objective of this study is thus to introduce a model framework to quantitatively detect changes in the soil water content response function and to classify the response as 'stable', 'resilient', or 'changed'.

In principle, it would be possible to develop a quantitative framework to detect changes in the response function exclusively based on experimental data of time series (without modelling) as for example by the application of wavelet analysis (Ehrhardt et al., 2025). In this study, however, we pursue a different approach based on predictive modelling, in which the temporal evolution of differences between measurements and predictions serves as an indicator of changes in the response function, yields both, accurate predictions of soil water content dynamics (not in the focus of this study) and detection of changes in the relationship between climate and soil water content dynamics.

# 2. Material and Methods

We developed a data-driven modelling framework that combines time-series decomposition of climatic inputs with a feed-forward neural network to predict the daily soil water content (Fig. 2).

Figure 2 Schematic overview of the modelling framework for daily soil water content prediction. Input features include the analysis of a climatic variable (top left), its long-term trend component (bottom left), and the categorical soil water content state ('wet', 'moderate', 'dry'; top middle). These features, derived from observed data, are used to train a feed-forward neural network (centre), which outputs daily predictions of volumetric water content (right). The model thus captures temporal soil water content dynamics based on structured climate signals and categorical conditions.

The approach explicitly incorporates precipitation, potential evapotranspiration, and their difference (climatic water balance) as primary inputs. Each of these climate drivers was decomposed into seasonal variations and long-term trend components using Seasonal-Trend decomposition (STL) and was included as a separate feature in the model (Boergens et al., 2024; Cleveland et al., 1990). The input features also included a categorical moisture class (type) that reflects the expected current soil water condition ('wet', 'moderate', 'dry'). This design reflects the understanding that changes in climate - such as shifts in rainfall and evaporative demand - substantially affect soil water availability and fluxes (Vereecken et al., 2022). The methodology is detailed in the following sections, which consists of five key steps: selecting study sites and datasets collected in contrasting hydro-climatic conditions (subsection 2.1), preprocessing the data and extracting meaningful signals features including STL (subsection 2.2), constructing and training a neural network model on a reference dataset (subsection 2.3), generating soil water content predictions for independent (non-training) sites using the trained


















163164





- model (subsection 2.4), evaluating the model's performance with statistical metrics (subsection 2.5),
- and physical consistency checks (subsection 2.6).

### 2.1 Study Sites and Data Selection

### 145 2.1.1 Lysimeter Network TERENO SOILCan

The study was conducted using lysimeter data from the TERENO-SOILCan lysimeter network in Germany (Pütz et al., 2016) with a focus on two locations: Bad Lauchstädt (BL) and Selhausen (SE) (Fig. 3). These sites were selected for their contrasting climatic regimes and the specific set-up of lysimeters, providing a natural experiment on how climate variability influences soil hydrological behaviour for a variety of soils. The TERENO-SOILCan lysimeters were moved between and within observatories according to a modified space-for-time approach, to expose them to different climates (Groh et al. 2020). This allows us to compare the ecosystem response of the same soil, but under different climatic conditions. Selhausen is characterized by a humid, Atlantic-influenced climate (annual precipitation around 720 mm and mean air temperature around 10 °C), whereas Bad Lauchstädt represents a drier, more continental climate (annual precipitation roughly 487 mm and mean air temperature approximately 8.8 °C); both climate descriptions are based on Pütz et al. (2016). Longterm observations confirm that Bad Lauchstädt experiences significantly lower rainfall and higher evaporative demand than Selhausen, yielding a higher aridity index (ratio of potential evapotranspiration to precipitation) and more pronounced dry spells in the growing season. By including both, a wetter site Selhausen and a drier site Bad Lauchstädt, the model is evaluated under distinctly different moisture regimes, which is critical for testing the generality of the approach and to separate between climatic and soil type effects. For each lysimeter station (Bad Lauchstädt and Selhausen), 12 lysimeters (1 m<sup>2</sup> surface area, 1.5 m depth) arranged in hexagons with 6 lysimeters around a service well were included in the analysis to monitor soil water content along with meteorological variables. In this study, lysimeters are not used for drainage or storage estimates, but rather as instruments providing long-term, high-resolution time series of soil water content and matric potential under field conditions. The lysimeters contain undisturbed soil columns collected at four different locations (see Fig. 3a), each with three replicates (Pütz et al., 2016) and were managed as

arable land under crop rotation. While fertilization practices differed regionally until spring 2019, the overall management concept was comparable, ensuring that differences in water dynamics can be attributed to changes in the climate and soil rather than the management (Pütz et al., 2016).

Figure 3 Overview of the study area with site locations, topsoil texture, and soil origin. (a) The TERENO-SOILCan network contains lysimeters from four climatic regions (different symbols and colours in map). Our analysis focuses on the TERENO-SOILCan sites Selhausen (SE) and Bad Lauchstädt (BL), located in the Eifel/lower rhine valley and Harz/central German lowland observatory of TERENO, respectively, because at both sites lysimeter clusters were built (represented by shaded areas and hexagons), collecting large soil columns from four distinct source regions (i.e., Dedelow (DD), Bad Laucstädt (BL), Sauerbach (SB), and Selhausen (SE)). (b) The analysed soil horizons (10 cm depth) cover two textural classes, shown in the USDA soil texture triangle, assigned to four different soil types and a range of soil organic carbon contents (SOC) (numbers in the legend).

To investigate the effects of climate on soil water dynamics, daily time series of precipitation (P), potential evapotranspiration (PET), matric potential, and volumetric soil water content (used as the target variable) were compiled. P at Selhausen was measured at the on-site SOILCan weather station, while for Bad Lauchstädt it was taken from the nearest long-term monitoring station operated by the Deutscher Wetterdienst (Leipzig/Halle, ID 2932; DWD Climate Data Center, 2025). PET was calculated with the FAO-56 Penman–Monteith model (Allen et al., 2006), using measured meteorological variables (air temperature, air pressure, relative humidity, radiation, and wind speed) according to SOILCan protocols (Pütz et al., 2016; Groh et al., 2020). In this context, the reference evapotranspiration (ET<sub>0</sub>) calculated with the Penman–Monteith model for a clipped grass surface (FAO-

56) is used as a proxy for potential evapotranspiration (PET), representing the site-independent atmospheric evaporative demand. Soil matric potential was measured using MPS-1 sensors (Decagon Devices Inc., Pullman, WA, USA), and volumetric soil water content was measured with time-domain reflectometry probes (CS610, Campbell Scientific, North Logan, UT, USA). The observational record spans the period from 2015 to 2023 and includes measurements taken at a depth of 10 cm (deeper soil layers were not analysed; Pütz et al., 2016).

#### 2.1.2 Definition of Reference Site for Model Framework

For model development, a single lysimeter moved from Dedelow to the Bad Lauchstädt lysimeter station (see Fig. 3a and 3b) was selected as the training dataset. This lysimeter was chosen due to its stable soil water content dynamics and minimal temporal drift in water retention properties over the observation period (see Fig. 4a). This lysimeter served as the reference dataset for developing the predictive model because it allows the definition of soil water content response function for the seasonal climatic conditions. The 23 remaining lysimeters at the Bad Lauchstädt and Selhausen site were used as independent test datasets (a contrasting example is shown in Fig. 4b) to evaluate model generalization and detect potential shifts in soil hydraulic behaviour across sites and years.

Figure 4 Soil water retention curves using data collected between 2015 and 2023 at 10 cm depth. Matric potential is plotted against volumetric water content, with data colour-coded by period: 2015–2016 (blue), 2017–2020 (green), and 2021–2023 (red). (a) Training lysimeter (moved from Dedelow to Bad Lauchstädt). (b) Test lysimeter (original soil from Selhausen in lysimeter station at Selhausen).









220221












233234


### 2.2 Data Preprocessing and Feature Engineering

All raw data were aggregated or resampled to a daily time step to support time-series analysis and

212 modelling. Any misaligned or duplicated timestamps were corrected to ensure consistency.

### 2.2.1 Seasonal-Trend decomposition

After cleaning and aligning the data, the climatic variables used as inputs in the modelling were selected. In addition to raw P and PET data, the daily climatic water balance (WB) was included as an explicit input. This variable reflects the net difference between P and PET, serving as a proxy for wetting or drying conditions. Positive values indicate potential moisture accumulation (e.g., during rainfalldominated periods), while negative values reflect high evaporative demand and drying conditions (e.g., during hot, dry days). Including the WB helps the model to distinguish humid periods from dry ones. By providing WB alongside P and PET, the model can learn both the individual and combined effects of P and evaporative demand on soil water content dynamics (Brocca et al., 2010; Uber et al., 2018). For example, it can infer that 10 mm of P during a high-PET summer day (low positive or negative WB) is less likely to increase soil water content than the same P on a cool, low-PET Day (high positive WB). To provide the model with structured representations of climate variability, each climatic time series was decomposed into additive components using STL based on LOESS with LOESS as acronym for 'Locally Estimated Scatterplot Smoothing' (Cleveland et al., 1990). STL is a non-parametric method that separates a time series into three interpretable parts: a seasonal component representing repeating seasonal patterns (such as wetting and drying cycles), a trend component capturing gradual long-term changes (such as climate shifts), and a residual component containing short-term irregularities and highfrequency noise (Cleveland et al., 1990). This decomposition was applied independently to the P, PET, and WB time series. Only the seasonal and trend components were retained as input features, as they contain meaningful patterns relevant to soil water content dynamics. The residual component, which lacks systematic structure, was excluded from further analysis. STL was configured with a cycle length of 180 days, representing the semi-annual wet-dry phases at the study sites. A LOESS smoother with a 90-day window was then applied to the de-seasonalized series to extract the trend component. This https://doi.org/10.5194/egusphere-2025-5141 Preprint. Discussion started: 24 October 2025 © Author(s) 2025. CC BY 4.0 License.

configuration was chosen to capture gradual, long-term changes in the climatic variables while reducing short-term fluctuations. Note, that near the ends of the time series the absence of future values causes the smoothing window to become asymmetric. As a result, the estimated trend becomes more sensitive to recent variability. This limitation does not affect the outcome of the analysis, as both the input features and the target variable (water content) are equally influenced by it. Each of the extracted seasonal and trend components from P, PET, and WB was included as input to the neural network alongside the original raw values. This allowed the model to learn structured seasonal behaviour—such as distinguishing the rising phase of spring wetting up of the soil profile from the declining phase of a summer dry-down—and to account for long-term shifts, such as gradual drying or changes in mean climate conditions.

### 2.2.2 Wetness classification

In addition to the continuous climate-related features, a categorical input was included to describe the soil's moisture condition as either 'dry', 'moderate', or 'wet'. These categories were defined using the soil water content time series from the training site, with thresholds based on quantiles of the full distribution. Specifically, values below the 30<sup>th</sup> percentile were labelled as 'dry', between the 30<sup>th</sup> and 70<sup>th</sup> percentiles as 'moderate', and above the 70<sup>th</sup> percentile as 'wet'. These categories were then encoded numerically prior to modelling, using values of 30 for 'dry', 20 for 'moderate', and 1 for 'wet'. This encoding allowed the categorical feature to be treated as ordinal variable and integrated into the neural network input layer alongside the other features. There are two reasons to include this feature. Firstly, the soil's current moisture condition can strongly influence its response to P and PET (Western & Grayson, 1998). For example, under dry conditions, more water can be absorbed by the soil due to its high storage capacity. In contrast, when soils are already wet or near saturation, infiltration capacity is reduced, and additional rainfall is more likely to result in runoff (Tromp-van Meerveld & McDonnell, 2006; Zehe & Blöschl, 2004). The second reason is the motivation to use remote sensing data in similar follow up studies, which are not yet accurate enough for modelling purposes but allow a general classification of the wetness status. Because (i) corresponding information on soil matric potential


























cannot be deduced at larger scale from remote sensing data and (ii) hysteresis in the soil water retention curve may lead to ambiguous thresholds, we focus here on soil water content measurements.

Another choice for the model framework that must be discussed is the choice of percentile thresholds. From a soil hydrological point of view, it would make sense that the thresholds defining the three classes 'wet', 'moderate' and 'dry' are chosen individually for each lysimeter (a wet clay soil may have very different water content values than a sandy soil). However, from a methodological point of view, we prefer to ensure that the model does not require a long time series to determine quantiles of soil water content data and that the model can be run solely based on the training site's distribution. Accordingly, the same percentile thresholds, derived from the training site, were applied to label daily water content values at the prediction sites. Note, that the application of the same percentile thresholds for all sites is not relevant for the detection of changes in the soil water content response function. Very similar results will be obtained for a site-specific percentile definition as shown in the supplementary material (section S1). After constructing all the above features, each daily input to the model consisted of (i) raw climate variables (P, PET, and WB), (ii) the STL-derived seasonal and trend components for each of those variables (six variables), and (iii) the categorical moisture label. All ten features were aligned by date to ensure consistency across inputs. This combination of raw values, decomposed temporal signals, and qualitative soil condition provides the model with a detailed daily representation of both external climatic forcing and internal system state.

### 2.3 Neural Network Architecture and Training

To model daily volumetric soil water contents, a feed-forward neural network was implemented. The architecture consisted of three hidden layers: two dense layers with 12 neurons each using ReLU activation functions (Rectified Linear Unit), followed by a batch normalization layer, and a third dense layer with 6 neurons. ReLU was chosen for its ability to introduce non-linearity while maintaining computational efficiency and avoiding vanishing gradient problems during training (Lu et al., 2020; Montesinos López et al., 2022). Batch normalization was applied to stabilize learning by reducing internal covariate shift, which improves convergence speed and training stability (Montesinos López et












300 301










311312



al., 2022). The output layer consisted of a single neuron with a linear activation function, which is standard for continuous regression tasks such as predicting soil water content.

The network was trained using input features derived from daily observations at the reference lysimeter at the Bad Lauchstädt site, covering the period 2015-2023. Prior to training, all continuous input features were standardized to have a mean of zero and a standard deviation of one using z-score normalization. The standardization parameters (mean and standard deviation) were computed solely from the training dataset and applied unchanged to the validation sets at the training site, as well as to the prediction sites, ensuring consistency across all data splits. The target variable, volumetric soil water content, was preserved in its original physical units (m<sup>3</sup> m<sup>-3</sup>), allowing for direct interpretation of the model outputs and associated errors in hydrologically meaningful terms. The model was compiled with the Adam optimizer, which adaptively adjusts learning rates and is widely used for its computational efficiency and stable convergence. Mean squared error (MSE) was used as the loss function due to its sensitivity to large deviations, making it suitable for continuous regression tasks. To monitor generalization, 30% of the data was withheld as a validation set and excluded from updating the weights between the nodes during training. The training procedure was initially set to proceed for a maximum of 1000 epochs. To prevent overfitting, an early stopping criterion was implemented based on validation loss. Specifically, training was terminated if no improvement in validation performance was observed over a predefined number of consecutive epochs (patience threshold). The model parameters from the epoch exhibiting the lowest validation loss were retained for final evaluation.

### 2.4 Testing the Neural Network

After training, the model was applied to the remaining 23 lysimeters across both Selhausen and Bad Lauchstädt, none of which were included in the training phase. All test inputs were processed using the same structure and normalization parameters derived from the training data. As outlined in Section 2.1, the experimental setup includes four soil types, each installed with three replicates at both sites (see Fig. 3). While the lysimeters at the Bad Lauchstädt lysimeter station share the same climatic setting as the training site, the lysimeters at Selhausen represent a more humid region. Accordingly, the raw data and seasonal trend data of the Selhausen climate were used as input for the prediction of soil water content

in lysimeters located at Selhausen. This configuration allows to evaluate (i) whether the soil water content response function determined for the training remains valid for different climates and soil types, and (ii) to detect potential temporal or structural changes in soil hydraulic behaviour. The evaluation and classification procedures are described in the following two subsections.

# 2.5 Detection of Change in Soil Water Content Response Function Based on Error Metrics

As explained in the introduction, we use error metrics to detect changes in the soil water content response function. While we use the Nash-Sutcliffe -Efficiency (NSE, see eq. 5) as a general descriptor of model error, we investigate temporal changes in model performance based on the Mean Bias (MB) that was calculated on an annual basis from 2015 to 2023. This year-by-year assessment does not rely on predefined change points and enables the detection of gradual or abrupt shifts in model performance directly from the data. MB measures the average signed difference between predicted and observed values, providing an estimate of systematic overestimation or underestimation over time (Moriasi et al., 2007; Liu et al., 2011) and is defined as:

$$MB = \frac{1}{N} \sum_{i=1}^{N} (\hat{\theta}_i - \theta_i) \tag{1}$$

where  $\hat{\theta}_i$  is the predicted volumetric water content at day i,  $\theta_i$  is the corresponding observation and N defines the number of available observations—prediction pairs. Although, the calculation uses daily values, MB is aggregated over yearly intervals to produce a single value per year, capturing annual patterns in prediction bias. Volumetric water contents (m³ m³) were multiplied by 100 prior to calculation, and MB is therefore reported in percentage (%). Positive MB values indicate systematic overestimation by the model, while negative values reflect underestimation. The annual assessment of MB allowed us to evaluate whether the soil water content response function remains consistent across time or shows temporal dynamics.

To classify the soil water content dynamics with respect to the resilience after the extreme summer 2018, we check if the deviation of the predictions based on a stable response function (developed with the training data) changes over the years. When the deviation in the first year (2015; i.e., before the

drought) is different from the deviation in the year 2023, we consider that the soil water content response function has changed (it is still possible that the response function may recover in the future) and the soil water content dynamics is classified accordingly as 'changed'. When the deviations at beginning and end are similar, but there was a period between 2018 and 2022 with a different deviation level, we conclude that the soil water content response function changed reversibly over time but recovered within the observation window and the lysimeter is classified as 'resilient'. The soil water content response is considered as 'stable' when the deviations remain similar during the entire observation period. As threshold we chose 1.52%, that equals the 3-fold of the standard deviation of the nine yearly MB values computed for the training site. The classification of the time series was thus expressed formally as:

351 'changed': 
$$|MB_{2023} - MB_{2015}| > 1.52\%$$
 (2)

352 'resilient': 
$$|MB_{2023} - MB_{2015}| \le 1.52\% \land |MB_{20xx} - MB_{2015}| > 1.52\%$$
 (3)

353 'stable': 
$$|MB_{2023} - MB_{2015}| \le 1.52\% \land |MB_{20xx} - MB_{2015}| \le 1.52\%$$
 (4)

with the logical operator Λ and the mean bias of a specific year with MB<sub>20xx</sub>, that shows the largest difference |MB<sub>year</sub>-MB<sub>2015</sub>| for the time period between 2018 and 2022 (starting with dry year 2018).

As general information on the different response function, we calculated the NSE coefficient (Moriasi et al., 2007; Nash & Sutcliffe, 1970). The NSE is a standard metric for hydrological model skill, with NSE = 1 indicating perfect agreement and NSE  $\leq$  0 stating that the model predictions are no better than using the mean of the observations. Mathematically, it is defined as:

360 
$$NSE = 1 - \frac{\sum_{i=1}^{N} (\theta_i - \hat{\theta}_i))^2}{\sum_{i=1}^{N} (\theta_i - \bar{\theta}))^2}$$
 (5)

where  $\hat{\theta}_i$  is the predicted volumetric water content at day i,  $\theta_i$  is the corresponding observed value,  $\bar{\theta}$  is the mean observed volumetric water content over the evaluation period, and N denotes the total number of valid data points used in the calculation. Following Moriasi et al. (2015), model performance was classified as very good for NSE > 0.80, good for  $0.70 

the soil hydraulic behaviour represented in the training data, potentially due to differences in soil properties or climate-induced structural changes.

### 2.6 Interpretation of Change in Response Function in Soil Physical Terms

The dynamics of MB (see above) was also used to assess changes in soil water retention curves (SWRCs), which were plotted for each test lysimeter on a yearly basis. As stated in eq. (1), a positive MB value corresponds to measured values that are smaller than the predictions. Because the predictions are based on the model trained for a specific lysimeter, we expect for a positive MB that the water content for the same environmental conditions (as manifested in the matric potential) is smaller in the test lysimeter compared to the lysimeter used for training (SWRC is shifted to the left). Analogously, for a consistently negative MB we expect that the test site retained more water at a given matric potential than the training site and the SWRC is shifted to the right. For a 'resilient' soil, the soil water retention curve will be shifted over time and will shift back close to the original position at the end of the observation period. Finally, for a soil with 'changed' response function, the water retention curve is drifting over time as well but without returning to its original position. In some cases, the temporal evolution of MB may not exactly follow the apparent shift of the SWRC, as additional vertical or slope changes could occur due to variations in porosity or pore-size distribution. These effects cannot be identified within the current framework but may contribute to deviations between MB dynamics and the apparent SWRC shift.

### 3. Results

Following the methodological framework described in Section 2.3-2.6, we present the results of model predictions across the test lysimeters to assess the resilience of the soil water content response function for the different lysimeters. We organize the section in four subsections according to the four different origins of the soil in the lysimeters (see Fig.3b) to discuss effects of soil origin and climatic conditions on the response function. In the last subsection (3.5) the results are summarized to allow direct comparison of all 24 lysimeters. Note, that all model results presented below are based on the soil water content classification ('wet', 'moderate', 'dry') as deduced from the lysimeter used for model training.

The corresponding figures using specific classification for each lysimeter are shown in supplementary information Fig. S4-S7.

### 3.1 Lysimeters with Same Soil as Used in Model Training (Dedelow Soils)

The neural network was trained to capture the soil water content response function of one lysimeter with sandy loam topsoil (Luvisol) extracted from Dedelow and translocated to the dry climatic region in Bad Lauchstädt (see Fig. S1 in supplementary material). The NSE of the training and validation of that specific lysimeter was very high with 0.91 indicating good model performance. The application of this response function to the other two lysimeters from Dedelow that were translocated to Bad Lauchstädt resulted in relatively high NSE values (0.79 and 0.84). However, the lower soil water contents observed during the summer of 2018 were not adequately captured as shown in Fig. 5a. More specifically, the time series show that predictions and observations matched closely in 2015, while after the dry summer of 2018 the model systematically overestimated water content in 2019 and 2020, before the agreement improved again towards the end of the period.

429

430

431

432

433

434

435

406 Figure 5 Analysis of soil water content dynamics (2015–2023) for a Dedelow-origin lysimeter tested at Bad 407 Lauchstädt. Panel (a) shows the time series of observed (blue) and predicted (orange) water content, with close 408 agreement in 2015, clear overestimation in 2019-2020 (predictions above observations), and improved 409 agreement again towards the end of the period. Panel (b) presents the temporal evolution of mean bias (MB), 410 remaining near zero until 2017, increasing to about 2-3 % in 2019-2020, and decreasing again to approximately 411 zero in 2022. Such soil water content response was classified as 'resilient'. Panel (c) displays soil water retention 412 curves from the training site (grey) and from selected years representing different MB conditions, with low-MB 413 years (2015, 2022) and high-MB years (2019, 2020). The curves are close to the training site in 2015, show a 414 shift to lower water contents in 2019–2020, and in 2022 return to the training site data. These changes are reflected in the MB development (Fig. 5b), with values increasing from near zero in 415 416 2015 to about 2-3% in 2019-2020 and then decreasing again towards 2022. The retention curves 417 confirm this interpretation (Fig. 5c). The year with low MB (2015) produced a SWRC close to the 418 measured curve of the training site, the years with high MB (2019-2020) were shifted to lower water 419 contents, and the later year with reduced MB (2022) returned to the measured SWRC of the training 420 site. Taken together, the time series, MB trend, and SWRCs show that the soil response was disturbed 421 after 2018 but later recovered, defining this lysimeter as 'resilient'. The same finding holds for the 422 simulations for the lysimeters translocated to Selhausen (less dry climate) with high NSE between 0.80-423 0.82. This indicates, that for this coarse soil (i) the effect of changing climatic conditions was rather 424 small (very good NSE classification for both sites) but (ii) that also these coarse textured topsoils do 425 not show identical response to the extreme year but each lysimeter reacts slightly different, indicating 426 slightly different structural properties.

3.2 Lysimeters with Soils Adapted to Climatic Conditions Similar with Those of the Model Training Site (Bad Lauchstädt soil)

The lysimeters filled with soil from Bad Lauchstädt (Chernozem) represent soils adapted to the climatic conditions under which the response function was calibrated. In case of dominant effect of climate on the soil water content response function, we could expect similar results as for the training lysimeter. For the soil remaining at the original site (Bad Lauchstädt), the model performance was very good (0.88–0.89). As shown for an example in Fig. 6a, the fit between observed and predicted water content was consistently close, with a tendency to slightly underestimate in the early years and to mildly overestimate after 2018, particularly in 2019–2020, before the agreement improved again in later years.

This is also manifested in the MB values that increased from slightly negative values in 2015 to about +1.5% in 2019, before decreasing again towards zero (Fig. 6b). The plotted SWRCs support this interpretation (Fig. 6c), with low MB years (2015 and 2016) showing a slight shift to higher water contents relative to the measured SWRC of the training site, and high-MB years (2019 and 2020) displaying a modest shift to lower water contents. Accordingly, the soil water content dynamics was classified as 'resilient'.

For the lysimeters transported from Bad Lauchstädt to Selhausen, the performance was more variable (NSE ranging from 0.50 to 0.84) corresponding to satisfactory to very good classifications, reflecting the stronger effect of the wetter climate. None of the three lysimeters who stayed in Bad Lauchstädt were classified as 'changed' but two out of three showed a systematic shift and were classified as 'changed' when translocated to Selhausen (see Table 1). In short, those examples show that the Bad Lauchstädt soil remained resilient under unchanged climate at Bad Lauchstädt but changed under the wetter climate at Selhausen.





455 456











over time (Fig. 7a).

Figure 6 Analysis of soil water content dynamics (2015–2023) for a Bad Lauchstädt-origin soil lysimeter tested at Bad Lauchstädt. (a) Comparison of measured (blue) and simulated (orange) daily water content values, showing high agreement in the early years and temporary overestimation in 2019-2020. (b) Mean Bias (MB) started slightly negative in 2015, increased to about +1.5 % in 2019, and then decreased again towards 2022. (c) Soil water retention curves (SWRCs) from the training site (grey) and from the same replicate for selected years with low MB (2015, 2016) and high MB (2019, 2020) show close agreement in the early years and a shift to lower water contents in 2019-2020. 3.3 Lysimeters with Soils Adapted to Climatic Conditions comparable with Those of the Model Training Site (Sauerbach soil) The findings are similar for the silt loam (Cambisol) from Sauerbach, representing soils adapted to climatic conditions comparable to those in Bad Lauchstädt. As in case of the soil from Bad Lauchstädt, soils from Sauerbach show higher NSE values when translocated to Bad Lauchstädt (0.81-0.88, very good) compared to those transferred to the wetter climate in Selhausen (0.74-0.79, good). This reflects, that the soil water content response function in the drier climate is not the same as in the wetter climate. In one illustrative case, the observed water content initially showed wetter dynamics than predicted, but gradually converged toward the model predictions by 2023, indicating a possible structural adjustment

Figure 7 Analysis of soil water content dynamics (2015–2023) for Sauerbach-origin lysimeter relocated to Selhausen (a) Comparison of observed and predicted daily volumetric water content (NSE = 0.74) After initial underestimation by the model, the observed and predicted values gradually converged, indicating a possible structural adjustment.(b) Temporal evolution of the mean bias (MB), which increased from about –5 % in 2015 to values close to zero by 2019–2023, consistent with the improved match between observed and predicted values shown in panel (a). (c) Soil water retention curves (SWRCs) from the training site (grey) and from selected years with low MB (2015, 2016) and high MB (2020, 2022) illustrate the same trend, with early years showing higher water contents at a given matric potential and later years shifting towards to the training curve.

This development is also evident in the MB values (Fig. 7b), which started strongly negative (-5%) in 2015–2016 and steadily increased toward values close to zero by 2023, indicating a progressive reduction of underestimation. The corresponding SWRCs (Fig. 7c) confirm this trend, with curves from early years (2015, 2016) showing higher water contents at a given matric potential compared to the measured SWRC of the training site and later years (2021, 2023) shifting closer to the reference, suggesting a gradual adjustment of hydraulic behaviour. In the case of soils from Sauerbach, there was a difference in quantification of resilience with respect to the classification of the soil water content used as input variable: with the classification based on the training lysimeter (with sandy loam in the




















504 505


topsoil), the soil water content dynamics was classified as 'changed' for all six lysimeters. But using the classification based on the soil water content statistics obtained for each lysimeter individually (see Fig. S6), the large water contents at the beginning were captured and only one lysimeter out of three was classified 'changed'. Independent of the water content classification, all lysimeters translocated to Selhausen were classified as 'changed', exhibiting the strongest response to relocation of all soils.

# 3.4 Lysimeters with Soils Adapted to Climatic Conditions Different from Those of the Model Training Site (Selhausen)

At last, we discuss the Selhausen silt loam (Luvisol), representing soils adapted to climatic conditions that were not included in the neural network training. The model performance was better for the replicates translocated to the drier Bad Lauchstädt climate (NSE = 0.86-0.92, very good), compared to slightly lower performance at their site of origin under humid Atlantic conditions (0.76-0.86, good to very good). The classification with respect to resilience helps to explain this, since Selhausen soils at their origin were mainly assigned to 'stable' or 'resilient' categories (see Table I), while the same soils translocated to Bad Lauchstädt showed a more variable pattern. This indicates, that the lower NSE at Selhausen does not represent a misfit of the model but reflects that the soils follow their own stable soil water content response function. One replicate at Selhausen (NSE = 0.85) reproduced the seasonal dynamics well, although differences between observed and predicted values remained visible in the wet season across several years (Fig. 8a). The MB shifted from negative values in the first years toward zero after 2019. Note, that an MB value of 0 does not mean that deviations disappeared, but that errors in wetter and drier phases compensated each other (Fig. 8b). The SWRCs were generally close to the training reference, but in later years small deviations appeared mainly at the saturated end (Fig. 8c). Overall, these changes remained below the assumed threshold, supporting a classification of the soil water content dynamics as 'stable'.

Figure 8 Soil water content dynamics (2015–2023) for a Selhausen-origin lysimeter tested at Selhausen. (a) Observed (blue) and predicted (orange) water content show fair agreement, with underestimation of water contents in the wet season. (b) Mean Bias (MB) fluctuated from negative values in the early years to values close to zero after 2019, but these variations remained below the threshold for change. (c) Soil water retention curves (SWRCs) from the training site (grey) and from selected years with low MB (2015, 2016) and higher MB (2019, 2020) reflect these minor variations, with the 2019 curve showing the strongest deviation yet remaining close to the training reference, consistent with the stable classification. The apparent cutoff at the wet end in (a) arises from the use of absolute rather than normalized values during training, as discussed in the Supplementary Material (Text S1).

# 3.5 Comparison of All Lysimeters

The comparison of the soil water content dynamics of all lysimeters indicate, that climatic shifts between sites - particularly between the continental Bad Lauchstädt and Atlantic-influenced Selhausen - can significantly alter the hydraulic response of the soil, even when texture remains constant. In general, prediction performance at Selhausen was lower, likely because the model was trained under the drier climate of Bad Lauchstädt, and therefore, failed to fully capture the soil—water interactions

emerging under wetter conditions (Fig. 9). The broader NSE range observed at Selhausen location further suggests increased structural variability among replicates.

Figure 9 Spread of Nash–Sutcliffe Efficiency (NSE) values across different soil origins and test locations. Each symbol represents one lysimeter from a given origin (x-axis) evaluated at Bad Lauchstädt or Selhausen (indicated by colour). The results highlight the influence of climate—soil interactions on model performance. Notably, Bad Lauchstädt-origin soils exhibited strong performance at their origin but a wider and lower range when tested at Selhausen, reflecting increased structural variability or climate-induced divergence in hydraulic response.

With respect to resilience of the soil water content response function, we show the temporal evolution of the mean bias for all lysimeters in Fig. 10 and summarize the results in Table 1. In Table 1 we add the general classification type ('stable', resilient' and 'changed') and calculate the average of 3 lysimeters (same material and same location) for the drift in mean bias value between the year 2015 and 2023 and the maximum deviation from year 2015 for the years between 2018 and 2022. The table shows that deviations from a 'stable' or 'resilient' response function mainly occur when soils from Dedelow, Bad Lauchstädt, and Sauerbach were translocated to Selhausen. Only in case of the soil from Selhausen, the response function remains 'stable'. It seems, that the soil material 'trained over decades' to the wetter climate in Selhausen adapts better to the extreme summer 2018.

Figure 10 Temporal evolution of Mean Bias (MB) for three representative lysimeter replicates, each classified into one of three structural response categories: (a) 'stable', (b) 'resilient', and (c) 'changed'. Thick dashed lines indicate the mean of the MB trend across all lysimeters within each classification group, with sample size (n) specified in the legend. Shaded areas represent  $\pm 1$  standard deviation. Thin grey lines show individual MB trajectories of the remaining lysimeters in each group. Highlighted blue lines depict selected replicates originating from and/or tested at distinct sites: (a)  $BL \rightarrow BL$  (soil material from Bad Lauchstädt tested at its origin), (b)  $DD \rightarrow BL$  (soil material from Dedelow tested at Bad Lauchstädt), and (c)  $BL \rightarrow SE$  (soil material from Bad Lauchstädt tested at Selhausen). These examples illustrate contrasting temporal patterns in structural response, ranging from sustained stability to progressive divergence from the trained site dynamics.

**Table 1:** Resilience of soil water content response function for the four soil materials translocated to Bad Lauchstädt and Selhausen. The 'type' describes the class of response function of the individual lysimeters (S for 'stable', R for 'resilient' and C for 'changed'). The 'drift' is the average value |MB<sub>2023</sub>-MB<sub>2015</sub>| of the three lysimeters with the difference in Mean Bias (MB) between years 2023 and 2015. The 'amplitude' is the maximum

difference of the Mean Bias between the first year (2015) and the years between 2018 and 2022 (denoted as year 20xx).

|                | Located at Bad Lauchstädt |       |           | Located at Selhausen                 |       |           |
|----------------|---------------------------|-------|-----------|--------------------------------------|-------|-----------|
|                | Type                      | Drift | Amplitude | Type                                 | Drift | Amplitude |
| Dedelow        | S,R,C                     | 1.09  | 1.76      | $\mathbb{R}, \mathbb{C}, \mathbb{C}$ | 1.72  | 2.11      |
| Bad Lauchstädt | $S,S,\mathbb{R}$          | 0.94  | 1.36      | $\mathbb{R}, \mathbb{C}, \mathbb{C}$ | 2.15  | 2.99      |
| Sauerbach      | C,C,C                     | 2.17  | 3.73      | C,C,C                                | 3.38  | 4.24      |
| Selhausen      | S.R.C                     | 0.96  | 2.24      | S.S.R                                | 0.47  | 1.78      |

# 4. Discussion

The results presented in Section 3 demonstrate that the model can reproduce soil water content dynamics reliably under stable conditions (as indicated by high NSE-values), but it exhibits limitations when soils undergo structural changes or are exposed to a different climate. Several soils showed a shift in the wet range, indicating that differences in soil water content response cannot be explained by texture alone but reflect the combined effects of climatic conditions and structural evolution. Based on these findings, the following discussion evaluates how assumptions of static hydraulic behaviour and response function affect model performance, examines the role of NSE and MB in identifying evolving system dynamics, and reflects on the broader implications for long-term modelling and soil water content monitoring.

### 4.1 Soil-Climate Interactions as Drivers of Hydraulic Response Function

The predictive success of data-driven models depends not only on the physical properties of soils but also on the climatic context in which those properties developed and continue to function. The present study shows that soils exhibit the most consistent replicate behaviour when evaluated under climate conditions similar to those of their origin, where gradual climatic changes over time have allowed their structure to adjust naturally. When exposed to faster or contrasting climatic shifts, as in translocated settings, the soil response becomes less predictable and less stable. This can be shown using the table S1 in the supplementary material file, which lists the average trends (difference in MB between year 2023 and 2015) and amplitudes (difference in MB between 2015 and the dry years) for three lysimeter replicates: only for the three lysimeters at the original locations Bad Lauchstädt or Selhausen), both



























of lysimeters. This suggests that the structure and function of the soil system cannot be meaningfully decoupled from its climatic history. Soils may develop pore arrangements, aggregation patterns, and, as a consequence, moisture retention characteristics that reflect long-term adaptation to local hydrological regimes. When these soils are translocated to environments with contrasting P and atmospheric demand (PET), their hydraulic response can shift in ways that are not captured by static texture-based estimates of soil hydraulic properties. Such context-dependent behaviour highlights the limitation of the common assumption that soils with the same texture will show comparable retention across regions, an assumption often made in the absence of better descriptors. Experimental evidence collected under natural conditions also indicates that this description is oversimplified (Hannes et al., 2016; Robinson et al., 2016; Aqel et al., 2024). In our case, even soils with similar textural composition exhibited different levels of model agreement depending on climate, highlighting that physical similarity (e.g. soil texture) does not guarantee functional equivalence in retention. For example, Selhausen-origin soils achieved higher NSE values when translocated to Bad Lauchstädt, likely because the model was trained under similar dry climatic conditions. However, classification results showed, that these soils retained greater structural stability at their origin, suggesting that predictive success under familiar climatic forcing does not necessarily imply hydraulic consistency. After the 2018 drought, the Selhausen soils translocated to Bad Lauchstädt converged toward similar dynamics across replicates, with MB stabilizing close to zero, indicating that their response functions adjusted consistently to the drier climate (see Fig. S2 in the supplementary material). However, a clear carry-over effect was observed: soil water in the upper 10 cm was not fully replenished during the wet phase of autumn and winter 2019 and only reached comparable, though slightly lower, values in winter 2020. This persistent deficit points to a structural legacy of the drought, where reduced pore connectivity and altered aggregation limited subsequent rewetting. A comparable multi-year legacy across the full soil column was reported in the TERENO-SOILCan lysimeter network by Groh et al. (2020).

drift and amplitude were below the stability threshold of 1.52% and can classified as 'stable' as group

All mentioned points underscore the importance of including very broad range of climatic forcing in the assessment of soil model transferability, as demonstrated by Groh et al. 2022. Our results also suggest that future efforts to generalize hydrological models should consider training under a range of climatic conditions to capture the full expression of soil—climate interactions, rather than relying on a single static representation. From a process-based perspective, these findings reflect that climate does not simply modulate soil water content inputs but actively shapes the retention and release behaviour of the soil pore network through structural evolution or breakdown. While management practices across sites were similar, minor differences in tillage and fertilization cannot be completely excluded and may have influenced soil structure and water retention. Nonetheless, the dominant control remains climatic forcing, which makes this consideration particularly relevant for climate-change experiments: models calibrated under past climatic conditions may not remain valid under the rapid climatic shifts projected for the coming decades. Neglecting this evolving soil—climate feedback could lead to substantial underestimation of future changes in soil hydraulic behaviour and associated ecosystem responses

### 4.2 High Predictive Performance Can Mask System Evolution

Although, several lysimeters achieved high predictive performance as expressed by high NSE values (Fig. 9), systematic trends in MB over time suggest that the underlying retention behaviour and soil water content response function may have shifted (Fig. 10 and Table 1). This was most apparent in Dedelow soils translocated to Bad Lauchstädt, where the model maintained high NSE values, but the MB increased across years (see Fig. S3). The corresponding shifts in the soil water retention curves confirmed a gradual change in how the soil retained water, despite the model continuing to predict moisture levels accurately.

This suggests that local structural changes can occur without immediate deterioration in model fit. The predictive framework remained effective in capturing the general moisture dynamics, but the relationship between matric potential and water content was no longer consistent with that observed during training. These findings highlight that high model accuracy does not guarantee stability in the hydraulic characteristic, particularly under changing environmental conditions. Identifying such divergence early is critical for maintaining reliable predictions in long-term monitoring.

### 4.3 Implications for Monitoring, Remote Sensing, and Soil Health

The classification outcomes across all lysimeters highlight the role of site memory and structural resilience in maintaining hydraulic behaviour under climatic stress. Soils assessed at their origin were more frequently classified as 'stable' or 'resilient' (e.g., Selhausen at Selhausen), while those translocated to different locations were more likely to be classified as 'changed' (e.g., Sauerbach at Bad Lauchstädt). These patterns indicate that soil structure, once adapted to specific climate regimes, may lose its functional integrity when exposed to new conditions. The presented methods allow us to detect emerging structural shifts that may be relevant for soil health assessment and could be used as indicator for deteriorated soil health status.

This has direct implications for long-term monitoring and remote sensing. Our model framework - by avoiding reliance on matric potential data and instead using moisture state categories and decomposed climatic features - is compatible with satellite-derived products. As remote sensing missions increasingly provide continuous global soil water content estimates, the proposed framework could be adapted for large-scale assessment of soil system stability. Furthermore, under scenarios of future climate change, where shifts in precipitation patterns and evaporative demand are expected, models trained on historical data may become progressively outdated. The presented residual-based approach (quantifying MB) enables early detection of such divergence, offering a method for identifying when re-training or reparameterization is needed to maintain predictive reliability under non-stationary conditions.

# 5. Summary and conclusions

The temporal variations in the water content of the topsoil define the amount of plant available water and oxygen supply, affecting ecosystem functions and soil health status. Reliable information on soil water content dynamics in response to atmospheric conditions is thus essential to detect and mitigate critical conditions. This response depends on soil hydraulic properties that are traditionally characterized by a time-invariant and unambiguous relationship between matric potential, water content, and hydraulic conductivity as deduced from small-scale lab experiments. In this study, we

























developed and applied a feed-forward neural network combined with seasonal trend analysis of climatic time series to quantify the soil water content response function after an extreme drought in summer 2018 in Germany. By analysing the time series of topsoil water content measured at two lysimeter stations of the TERENO SOILCan network, we summarize the conclusion on the soil water content response function as follows: 50% of the lysimeters showed changes in soil water content dynamics after the dry summer 2018. The other half showed a resilient behaviour, and the soil water content response function was not permanently changed. The changes in soil water content response function were manifested as (i) temporal trends in prediction error (mean bias) and (ii) shifts in the soil water characteristics function. The soil water content response function is adapted to climatic conditions as manifested by (i) smallest changes in lysimeters that were not translocated and (ii) decreased model performance for applications of a response function that was determined for another climate Good model performance as expressed by high Nash-Shutcliff-Efficiency values does not correspond to stable soil water content response function that was only detected by temporal trends in error metrics The study revealed that extreme climatic events can permanently change the soil hydraulic properties, but the lack of resilience depends on the soil and the climatic conditions. We argue that the response depends on the range of climatic conditions experienced in the past that allowed adaptation of soil structural properties. Because the presented model framework (i) does not aim to predict successfully time series in water content and (ii) does only require categorical water content information ('stable', 'resilient', 'changed'), it can be applied to larger scale using remote sensing data that do not provide accurate soil water content values but reliable trends, enabling to detect changes in hydraulic behaviour at the ecosystem scale.

Author contributions 681 NA, AC, and PL designed the study. NA and PL conducted the research. NA developed the model and 682 wrote the code. NA and PL prepared the manuscript with contributions of all co-authors. JG and RG 683 provided and quality-controlled the lysimeter data. 684 Acknowledgements 685 We acknowledge the support of the TERENO-SOILCan network that were funded by the Helmholtz 686 687 Association (HGF) and the Federal Ministry of Education and Research (BMBF). We thank Ferdinand Engels, Robert Lüdtke, Werner Küpper, Ines Merbach, Philipp Meulendick, and Syliva Schmögner for 688 the instrument operation and data processing at both sites. We also thank Hans-Jörg Vogel for his 689 constructive feedback and insightful comments on the manuscript. Nedal Aqel acknowledges the 690 691 utilization of ChatGPT to enhance coherence within certain sections of the paper. Financial support 692 693 This research is part of the project AI4SoilHealth of the European Union's Horizon research and 694 innovation programme (grant agreement No. 101086179). This work has received funding from the 695 Swiss State Secretariat for Education, Research and Innovation (SERI).

31

| 697                             | References                                                                                                                                                                                                                                                                                                                                                                                                                                           |
|---------------------------------|------------------------------------------------------------------------------------------------------------------------------------------------------------------------------------------------------------------------------------------------------------------------------------------------------------------------------------------------------------------------------------------------------------------------------------------------------|
| 698<br>699<br>700               | Allen, R. G., Pereira, L. S., Raes, D., and Smith, M.: Crop evapotranspiration – Guidelines for computing crop water requirements, FAO Irrigation and Drainage Paper 56, Food and Agriculture Organization of the United Nations, Rome, 300 pp., 2006.                                                                                                                                                                                               |
| 701<br>702<br>703<br>704        | Aqel, N., Reusser, L., Margreth, S., Carminati, A., and Lehmann, P.: Prediction of hysteretic matric potential dynamics using artificial intelligence: application of autoencoder neural networks, Geosci. Model Dev., 17, 6949–6966, https://doi.org/10.5194/gmd-17-6949-2024, 2024.                                                                                                                                                                |
| 705<br>706<br>707<br>708<br>709 | Blöschl, G., Bierkens, M. F. P., Chambel, A., Cudennec, C., Destouni, G., Fiori, A., Kirchner, J. W., McDonnell, J. J., Savenije, H. H. G., Sivapalan, M., Stumpp, C., Toth, E., Volpi, E., Carr, G., Lupton, C., Salinas, J., Széles, B., Viglione, A., Aksoy, H., and Zhang, Y.: Twenty-three unsolved problems in hydrology (UPH) – a community perspective, Hydrol. Sci. J., 64, 1141–1158, https://doi.org/10.1080/02626667.2019.1620507, 2019. |
| 710<br>711<br>712               | Boergens, E., Güntner, A., Sips, M., Schwatke, C., and Dobslaw, H.: Interannual variations of terrestrial water storage in the East African Rift region, Hydrol. Earth Syst. Sci., 28, 4733–4754, https://doi.org/10.5194/hess-28-4733-2024, 2024.                                                                                                                                                                                                   |
| 713<br>714<br>715<br>716        | Bogena, H. R., Huisman, J. A., Güntner, A., Hübner, C., Kusche, J., Jonard, F., Vey, S., and Vereecken, H.: Emerging methods for noninvasive sensing of soil moisture dynamics from field to catchment scale: a review, WIREs Water, 2, 635–647, https://doi.org/10.1002/wat2.1097, 2015.                                                                                                                                                            |
| 717<br>718<br>719               | Brocca, L., Melone, F., Moramarco, T., Wagner, W., Naeimi, V., Bartalis, Z., and Hasenauer, S.: Improving runoff prediction through the assimilation of the ASCAT soil moisture product, Hydrol. Earth Syst. Sci., 14, 1881–1893, https://doi.org/10.5194/hess-14-1881-2010, 2010.                                                                                                                                                                   |
| 720<br>721                      | Cleveland, R. B., Cleveland, W. S., McRae, J. E., and Terpenning, I.: STL: A seasonal-trend decomposition procedure based on Loess, J. Off. Stat., 6, 3–73, 1990.                                                                                                                                                                                                                                                                                    |
| 722<br>723<br>724               | Detty, J. M. and McGuire, K. J.: Threshold changes in storm runoff generation at a till-mantled headwater catchment, Water Resour. Res., 46, W07525, https://doi.org/10.1029/2009WR008102, 2010.                                                                                                                                                                                                                                                     |
| 725<br>726<br>727               | Deutscher Wetterdienst (DWD) Climate Data Center (CDC): Historical daily precipitation data, station Leipzig/Halle (ID 2932), available at: https://www.dwd.de/cdc (last access: 28 August 2025), 2025.                                                                                                                                                                                                                                              |
| 728<br>729<br>730               | Ehrhardt, A., Groh, J., and Gerke, H. H.: Effects of different climatic conditions on soil water storage patterns, Hydrol. Earth Syst. Sci., 29, 313–334, https://doi.org/10.5194/hess-29-313-2025, 2025.                                                                                                                                                                                                                                            |
| 731<br>732<br>733               | Fatichi, S., Or, D., Walko, R., Vereecken, H., Young, M. H., Ghezzehei, T. A., Hengl, T., Kollet, S., Agam, N., and Avissar, R.: Soil structure is an important omission in Earth system models, Nat. Commun., 11, 522, https://doi.org/10.1038/s41467-020-14411-z, 2020.                                                                                                                                                                            |
| 734<br>735<br>736               | Fu, Z., Hu, W., Beare, M., Thomas, S., Carrick, S., Dando, J., Langer, S., Müller, K., Baird, D., and Lilburne, L.: Land use effects on soil hydraulic properties and the contribution of soil organic carbon, J. Hydrol., 602, 126741, https://doi.org/10.1016/j.jhydrol.2021.126741, 2021.                                                                                                                                                         |

737 Groh, J., Diamantopoulos, E., Duan, X., Ewert, F., Heinlein, F., Herbst, M., Holbak, M., Kamali, B., 738 Kersebaum, K.-C., Kuhnert, M., Nendel, C., Priesack, E., Steidl, J., Sommer, M., Pütz, T., 739 Vanderborght, J., Vereecken, H., Wallor, E., Weber, T. K. D., and Gerke, H. H.: Same soil, 740 different climate: crop model intercomparison on translocated lysimeters, Vadose Zone J., 21, 741 e20202, https://doi.org/10.1002/vzj2.20202, 2022. 742 Groh, J., Vanderborght, J., Pütz, T., Vogel, H.-J., Gründling, R., Rupp, H., Rahmati, M., Sommer, 743 M., Vereecken, H., and Gerke, H. H.: Responses of soil water storage and crop water use 744 efficiency to changing climatic conditions: a lysimeter-based space-for-time approach, Hydrol. 745 Earth Syst. Sci., 24, 1211-1225, https://doi.org/10.5194/hess-24-1211-2020, 2020. 746 Hannes, M., Wollschläger, U., Wöhling, T., and Vogel, H.-J.: Revisiting hydraulic hysteresis based 747 on long-term monitoring of hydraulic states in lysimeters, Water Resour. Res., 52, 3847-3865, https://doi.org/10.1002/2015WR018319, 2016. 748 749 Hari, V., Rakovec, O., Markonis, Y., Hanel, M., and Kumar, R.: Increased future occurrences of 750 the exceptional 2018-2019 Central European drought under global warming, Sci. Rep., 10, 751 12207, https://doi.org/10.1038/s41598-020-68872-9, 2020. 752 Herbrich, M. and Gerke, H. H.: Scales of water retention dynamics observed in eroded Luvisols 753 from an arable postglacial soil landscape, Vadose Zone J., 16, 1–12, 754 https://doi.org/10.2136/vzj2017.01.0003, 2017. 755 Hrachowitz, M., Savenije, H. H. G., Blöschl, G., McDonnell, J. J., Sivapalan, M., Pomeroy, J. W., 756 Arheimer, B., Blume, T., Clark, M. P., Ehret, U., Fenicia, F., Freer, J. E., Gelfan, A., Gupta, H. V., 757 Hughes, D. A., Hut, R. W., Montanari, A., Pande, S., Tetzlaff, D., and Cudennec, C.: A decade of 758 Predictions in Ungauged Basins (PUB) – a review, Hydrol. Sci. J., 58, 1198–1255, 759 https://doi.org/10.1080/02626667.2013.803183, 2013. 760 Jarvis, N. J., Beven, K. J., Larsbo, M., van Genuchten, M. T., Vereecken, H., and Vogel, H.-J.: 761 Interactions between soil structure dynamics, hydrological processes and organic matter 762 cycling: a new soil-crop model, Eur. J. Soil Sci., 75, e13455, https://doi.org/10.1111/ejss.13455, 763 2024. 764 Kratzert, F., Klotz, D., Shalev, G., Klambauer, G., Hochreiter, S., and Nearing, G. S.: Towards 765 learning universal, regional, and local hydrological behaviours via machine learning applied to 766 large-sample datasets, Hydrol. Earth Syst. Sci., 23, 5089-5110, https://doi.org/10.5194/hess-767 23-5089-2019, 2019. 768 Lehmann, P., Bickel, S., Wei, Z., and Or, D.: Physical constraints for improved soil hydraulic 769 parameter estimation by pedotransfer functions, Water Resour. Res., 56, e2019WR025963, 770 https://doi.org/10.1029/2019WR025963, 2020. 771 Liu, J., Hughes, D., Rahmani, F., Lawson, K., and Shen, C.: Evaluating a global soil moisture 772 dataset from a multitask model (GSM3 v1.0) with potential applications for crop threats, 773 Geosci. Model Dev., 16, 1553–1567, https://doi.org/10.5194/gmd-16-1553-2023, 2023. 774 Liu, J., Rahmani, F., Lawson, K., and Shen, C.: A multiscale deep learning model for soil moisture 775 integrating satellite and in situ data, Geophys. Res. Lett., 49, e2021GL096847, https://doi.org/10.1029/2021GL096847, 2022. 776 777 Liu, Y. Y., Parinussa, R. M., Dorigo, W. A., De Jeu, R. A. M., Wagner, W., van Dijk, A. I. J. M.,

McCabe, M. F., and Evans, J. P.: Developing an improved soil moisture dataset by blending

780 https://doi.org/10.5194/hess-15-425-2011, 2011. 781 Lu, L., Shin, Y., Su, Y., and Karniadakis, G. E.: Dying ReLU and initialization: theory and numerical 782 examples, Commun. Comput. Phys., 28, 1671-1706, https://doi.org/10.4208/cicp.OA-2020-783 0165, 2020. 784 Melsen, L. A. and Guse, B.: Climate change impacts model parameter sensitivity - implications 785 for calibration strategy and model diagnostic evaluation, Hydrol. Earth Syst. Sci., 25, 1307-786 1332, https://doi.org/10.5194/hess-25-1307-2021, 2021. 787 Milly, P. C. D., Betancourt, J., Falkenmark, M., Hirsch, R. M., Kundzewicz, Z. W., Lettenmaier, D. 788 P., and Stouffer, R. J.: Stationarity is dead: whither water management?, Science, 319, 573-574, https://doi.org/10.1126/science.1151915, 2008. 789 790 Montanari, A., Young, G., Savenije, H. H. G., Hughes, D., Wagener, T., Ren, L.-L., Koutsoyiannis, 791 D., Cudennec, C., Toth, E., Grimaldi, S., Blöschl, G., Sivapalan, M., Beven, K., Gupta, H., Hipsey, 792 B., Schaefli, B., Arheimer, B., Boegh, E., Schymanski, S. J., and Belyaev, V.: "Panta Rhei-793 Everything Flows": change in hydrology and society—The IAHS Scientific Decade 2013–2022, 794 Hydrol. Sci. J., 58, 1256-1275, https://doi.org/10.1080/02626667.2013.809088, 2013. 795 Montesinos López, O. A., Montesinos López, A., and Crossa, J.: Fundamentals of artificial neural 796 networks and deep learning, in: Multivariate Statistical Machine Learning Methods for 797 Genomic Prediction, Springer International Publishing, Cham, 379-425, 798 https://doi.org/10.1007/978-3-030-89010-0 10, 2022. 799 Moriasi, D. N., Arnold, J. G., Van Liew, M. W., Bingner, R. L., Harmel, R. D., and Veith, T. L.: Model evaluation guidelines for systematic quantification of accuracy in watershed 800 801 simulations, Trans. ASABE, 50, 885–900, https://doi.org/10.13031/2013.23153, 2007. 802 Moriasi, D. N., Gitau, M. W., Pai, N., and Daggupati, P.: Hydrologic and water quality models: 803 Performance measures and evaluation criteria. Trans. ASABE. 58, 1763–1785, 2015. https://web.ics.purdue.edu/~mgitau/pdf/Moriasi%20et%20al%202015.pdf 804 805 Mosavi, A., Ozturk, P., and Chau, K.-W.: Flood prediction using machine learning models: 806 literature review, Water, 10, 1536, https://doi.org/10.3390/w10111536, 2018. 807 Nash, J. E. and Sutcliffe, J. V.: River flow forecasting through conceptual models part I - A 808 discussion of principles, J. Hydrol., 10, 282-290, https://doi.org/10.1016/0022-1694(70)90255-809 6, 1970. 810 Nearing, G. S., Kratzert, F., Sampson, A. K., Pelissier, C. S., Klotz, D., Frame, J. M., Prieto, C., and 811 Gupta, H. V.: What role does hydrological science play in the age of machine learning?, Water Resour. Res., 57, e2020WR028091, https://doi.org/10.1029/2020WR028091, 2021. 812 813 O., S. and Orth, R.: Global soil moisture data derived through machine learning trained with in-814 situ measurements, Sci. Data, 8, 170, https://doi.org/10.1038/s41597-021-00964-1, 2021. 815 Pütz, T., Kiese, R., Wollschläger, U., Groh, J., Rupp, H., Zacharias, S., Priesack, E., Gerke, H. H., 816 Gasche, R., Bens, O., Borg, E., Baessler, C., Kaiser, K., Herbrich, M., Munch, J.-C., Sommer, M., 817 Vogel, H.-J., Vanderborght, J., and Vereecken, H.: TERENO-SOILCan: a lysimeter-network in 818 Germany observing soil processes and plant diversity influenced by climate change, Environ. 819 Earth Sci., 75, 1242, https://doi.org/10.1007/s12665-016-6031-5, 2016.

passive and active microwave satellite-based retrievals, Hydrol. Earth Syst. Sci., 15, 425-436,

| 820<br>821<br>822               | Reichstein, M., Camps-Valls, G., Stevens, B., Jung, M., Denzler, J., Carvalhais, N., and Prabhat: Deep learning and process understanding for data-driven Earth system science, Nature, 566, 195–204, https://doi.org/10.1038/s41586-019-0912-1, 2019.                                                                                                                                             |
|---------------------------------|----------------------------------------------------------------------------------------------------------------------------------------------------------------------------------------------------------------------------------------------------------------------------------------------------------------------------------------------------------------------------------------------------|
| 823<br>824<br>825               | Robinson, D. A., Jones, S. B., Lebron, I., Reinsch, S., Domínguez, M. T., Smith, A. R., Jones, D. L., Marshall, M. R., and Emmett, B. A.: Experimental evidence for drought induced alternative stable states of soil moisture, Sci. Rep., 6, 20018, https://doi.org/10.1038/srep20018, 2016.                                                                                                      |
| 826<br>827<br>828               | Robinson, D. A., Nemes, A., Reinsch, S., Radbourne, A., Bentley, L., and Keith, A. M.: Global meta-analysis of soil hydraulic properties on the same soils with differing land use, Sci. Total Environ., 852, 158506, https://doi.org/10.1016/j.scitotenv.2022.158506, 2022.                                                                                                                       |
| 829<br>830<br>831               | Seneviratne, S. I., Corti, T., Davin, E. L., Hirschi, M., Jaeger, E. B., Lehner, I., Orlowsky, B., and Teuling, A. J.: Investigating soil moisture—climate interactions in a changing climate: a review, Earth-Sci. Rev., 99, 125–161, https://doi.org/10.1016/j.earscirev.2010.02.004, 2010.                                                                                                      |
| 832<br>833<br>834<br>835        | Shen, C., Laloy, E., Elshorbagy, A., Albert, A., Bales, J., Chang, FJ., Ganguly, S., Hsu, KL., Kifer, D., Fang, Z., Fang, K., Li, D., Li, X., and Tsai, WP.: HESS Opinions: Incubating deep-learning-powered hydrologic science advances as a community, Hydrol. Earth Syst. Sci., 22, 5639–5656, https://doi.org/10.5194/hess-22-5639-2018, 2018.                                                 |
| 836<br>837<br>838               | Sun, W., Zhou, S., Yu, B., Zhang, Y., Keenan, T., and Fu, B.: Soil moisture—atmosphere interactions drive terrestrial carbon—water trade-offs, Commun. Earth Environ., 6, 169, https://doi.org/10.1038/s43247-025-02145-z, 2025.                                                                                                                                                                   |
| 839<br>840<br>841               | Tromp-van Meerveld, H. J. and McDonnell, J. J.: Threshold relations in subsurface stormflow: 1. A 147-storm analysis of the Panola hillslope, Water Resour. Res., 42, W02410, https://doi.org/10.1029/2004WR003778, 2006.                                                                                                                                                                          |
| 842<br>843<br>844<br>845        | Uber, M., Vandervaere, JP., Zin, I., Braud, I., Heistermann, M., Legoût, C., Molinié, G., and Nord, G.: How does initial soil moisture influence the hydrological response? A case study from southern France, Hydrol. Earth Syst. Sci., 22, 6127–6146, https://doi.org/10.5194/hess-22-6127-2018, 2018.                                                                                           |
| 846<br>847<br>848               | Vaze, J., Post, D. A., Chiew, F. H. S., Perraud, JM., Viney, N. R., and Teng, J.: Climate non-stationarity – validity of calibrated rainfall–runoff models for use in climate change studies, J. Hydrol., 394, 447–457, https://doi.org/10.1016/j.jhydrol.2010.09.018, 2010.                                                                                                                       |
| 849<br>850<br>851<br>852<br>853 | Vereecken, H., Amelung, W., Bauke, S. L., Bogena, H., Brüggemann, N., Montzka, C., Vanderborght, J., Bechtold, M., Blöschl, G., Carminati, A., Javaux, M., Konings, A. G., Kusche, J., Neuweiler, I., Or, D., Steele-Dunne, S., Verhoef, A., Young, M., and Zhang, Y.: Soil hydrology in the Earth system, Nat. Rev. Earth Environ., 3, 573–587, https://doi.org/10.1038/s43017-022-00324-6, 2022. |
| 854<br>855<br>856               | Vereecken, H., Huisman, J. A., Bogena, H., Vanderborght, J., Vrugt, J. A., and Hopmans, J. W.: On the value of soil moisture measurements in vadose zone hydrology: a review, Water Resour. Res., 44, W00D06, https://doi.org/10.1029/2008WR006829, 2008.                                                                                                                                          |
| 857<br>858<br>859               | Wankmüller, F. J. P., Delval, L., Lehmann, P., Baur, M. J., Cecere, A., Wolf, S., Or, D., Javaux, M., and Carminati, A.: Global influence of soil texture on ecosystem water limitation, Nature, 635, 631–638, https://doi.org/10.1038/s41586-024-08089-2, 2024.                                                                                                                                   |

https://doi.org/10.5194/egusphere-2025-5141 Preprint. Discussion started: 24 October 2025 © Author(s) 2025. CC BY 4.0 License.

| 360<br>361<br>362 | Western, A. W. and Grayson, R. B.: The Tarrawarra data set: soil moisture patterns, soil characteristics, and hydrological flux measurements, Water Resour. Res., 34, 2765–2768, https://doi.org/10.1029/98WR01833, 1998.                                                |
|-------------------|--------------------------------------------------------------------------------------------------------------------------------------------------------------------------------------------------------------------------------------------------------------------------|
| 363<br>364<br>365 | Xoplaki, E., Ellsäßer, F., Grieger, J., Nissen, K. M., Pinto, J. G., Augenstein, M., Chen, TC., and Wolf, F.: Compound events in Germany in 2018: drivers and case studies, Nat. Hazards Earth Syst. Sci., 25, 541–564, https://doi.org/10.5194/nhess-25-541-2025, 2025. |
| 366<br>367<br>368 | Zehe, E. and Blöschl, G.: Predictability of hydrologic response at the plot and catchment scales: role of initial conditions, Water Resour. Res., 40, W10240, https://doi.org/10.1029/2003WR002869, 2004.                                                                |
| 369<br>370<br>371 | Šimůnek, J., van Genuchten, M. T., and Šejna, M.: Recent developments and applications of the HYDRUS computer software packages, Vadose Zone J., 15, 1–25, https://doi.org/10.2136/vzj2016.04.0033, 2016.                                                                |
| 372               |                                                                                                                                                                                                                                                                          |

36