# Peer review of "Detecting the resilience of soil moisture dynamics to drought periods as function of soil type and climatic region"

_EGUsphere, 2025_

## Author Comment (AC1)

**REVIEW 1 (our reply in black)**

This paper applies statistical modelling techniques (combining a neural network model with seasonal trend analysis) to a comprehensive dataset of soil water contents and pressure potentials measured at 10 cm depth in lysimeters moved to two different locations, in order to identify shifts in hydrological responses to climate forcing.

The shifts in these "in situ" water retention curves (WRC) are intriguing and really quite dramatic (e.g. figures 4, 7 and 8). But I do wonder about the mechanisms and underlying processes. The authors are rather vague about the causes, suggesting that they are due to changes in soil structure tiggered by climate (lines 674-676). I'm not fully convinced about this interpretation, not least because the largest changes seem to occur in the very dry range of the WRC where structure should not play such a large role.

We thank the reviewer for raising this important point regarding the interpretation of the observed shifts in the in-situ water retention curves. We agree that our original wording could be interpreted as implying that soil structural change is a verified causal mechanism. This was not our intention.

In the manuscript, soil structure was presented as a plausible explanation based on established soil physics understanding that drought can alter pore networks and hydraulic behaviour. However, as in our response to Reviewer 2, we emphasize that the present study does not directly measure pore geometry, aggregation, wettability, or contact-angle effects. Therefore, we cannot verify specific physical mechanisms responsible for the observed SWRC shifts. Instead, the model-based analysis identifies persistent changes in the soil water content response function empirically, while the discussion of soil structural or pore-scale processes serves as a theoretical interpretation consistent with existing literature.

Regarding your comment on the relevance of soil structural effects in the dry range, the effect could be larger than expected, considering that aggregation at various scales and contact networks may change considerably by shrinkage during drying (and following swelling). Such effects (as for example measured by Berli et al., 2008 Water Resour. Res., 44, W00C09) may also explain the huge variability of unsaturated hydraulic properties predicted by pedotransfer functions in the dry range. We have added a sentence in the Introduction to clarify that.

Following the reviewer's guidance, we have revised the manuscript to clarify that references to soil structural or pore-scale change are presented as possible mechanisms rather than confirmed causal evidence. The revised text now states that the observed SWRC shifts reflect changes in soil hydraulic behaviour inferred from the data, while structural and interfacial processes are discussed as plausible explanations that require targeted future measurements for verification.

This revision aligns with the general modelling philosophy of the study, where deviations between observed and simulated behaviour are used to detect change in the soil–climate response function, while mechanistic explanations remain interpretative rather than experimentally proven.

We will replace "structural variability" with "variability in hydraulic response".

We will replace "structural changes" with "persistent changes in hydraulic behaviour" and "structural evolution" with "evolving soil hydraulic response".

(In this respect, I think the WRC curves should be plotted with matric potential on a log-axis for improved readability. On a linear scale, we can't really see what is happening close to saturation, which is where most of the structural changes would be expected).

We agree that logarithmic scaling of matric potential is commonly used to improve visualization of the near-saturated range of water retention curves.

However, in this study the purpose of presenting the SWRCs is not to analyse pore-scale behaviour close to saturation or to diagnose specific structural mechanisms. Rather, the curves are shown to demonstrate that the bias drift detected by the model corresponds to a **real, physically observable shift in the soil water retention relationship itself**, across the full range of observed conditions.

For this purpose, a linear scale is intentionally used to allow direct visual comparison of curve displacement between years and between lysimeters, and to highlight the magnitude and direction of shifts consistently with the mean-bias analysis. Using a logarithmic axis would compress large parts of the potential range and reduce the visual clarity of the overall curve displacement that is central to our change-detection framework.

We therefore prefer to retain the linear representation, as it best supports the methodological objective of linking model-detected drift to observable shifts in the empirical SWRC.

There may be alternative explanations for the observations, including (slowly reversible) swell-shrink behaviour and preferential (non-equilibrium) flow. I would encourage the authors to try to strengthen the discussion and interpretation of the data with respect to the underlying mechanisms, including the above-mentioned processes. Nevertheless, although the responses to climate of apparent WRC observed by the authors "in situ" seem stronger than I would expect (especially in the dry range), I am aware of two previous large-scale (regional-continental) statistical analyses of water retention curves measured in the laboratory that have shown significant impacts of climatic factors on the structural pore space (Hirmas, D. et al. 2018, Nature 561, 100-103; Klöffel, T., et al., 2024. Geoderma, 442, 116772). These studies could be mentioned as they would give support to the authors' inferences and interpretations.

We agree that several pore-scale and hydraulic processes could contribute to the observed shifts in the apparent water retention curves, including slowly reversible swell–shrink behaviour and preferential (non-equilibrium) flow. As also noted in our response to Reviewer 2, the present study does not directly measure pore geometry, aggregate dynamics, or flow regime, and therefore cannot isolate or verify specific physical mechanisms. Our discussion of soil structural or pore-scale processes is thus intended as theoretical interpretation rather than confirmed causal evidence.

Following the reviewer's suggestion, we have strengthened the discussion by explicitly acknowledging alternative processes that may contribute to the observed behaviour, including swell–shrink dynamics and non-equilibrium flow effects. In addition, we have incorporated the two large-scale studies highlighted by the reviewer, which demonstrate statistically significant links between climatic factors and structural pore-space organization across regional to continental scales. These studies provide independent support for our interpretation that climatic forcing can influence soil hydraulic behaviour and apparent water retention characteristics.

We add this line 604:

"This suggests that the structure and function of the soil system cannot be meaningfully decoupled from its climatic history……

Alternative pore-scale processes, including slowly reversible swell–shrink behaviour and non-equilibrium preferential flow, may also contribute to the observed shifts in apparent SWRCs behaviour. Recent large-scale analyses have demonstrated statistically significant impacts of climatic factors on soil structural pore space and laboratory-measured retention characteristics (Hirmas et al., 2018; Klöffel et al., 2024), providing independent support for the plausibility of climate-induced modification of soil hydraulic response."

Specific comments

1. Line 40: "indicator" rather than "factor"?

Thanks, we changed it accordingly.

2. Lines 40-41: "Dominant"? This seems a bit far-fetched. I don't think soil water content at 10 cm depth is the best hydrological indicator of soil health. It could certainly indirectly reflect risks of surface runoff, but then surely surface runoff itself would be a much better indicator of soil health? Contrary to claims made in the paper, I don't think water content so close to the soil surface says so much about the supply of water to plants (rooting depth and water availability in deeper soil layers will be far more important). I think the authors should tone down the emphasis in the introduction of the relevance of this kind of study to soil health.

We agree that soil water content at 10 cm depth should not be interpreted as a direct indicator of plant water availability or overall soil health, as root-zone water supply and deeper soil layers play a more dominant role in these processes. Our intention was not to claim that near-surface soil moisture alone defines soil health, but rather that it reflects near-surface hydraulic conditions influencing infiltration, evaporation, and the initiation of surface runoff.

Following the reviewer's guidance, we have toned down the wording in the Introduction. We now avoid referring to near-surface soil water content as a dominant indicator of soil health or plant water supply and instead describe it as an indicator of near-surface hydraulic functioning and land–atmosphere exchange processes, which are relevant for several soil health–related surface processes. This revision better aligns the text with the actual scope of our measurements and avoids overstatement of broader ecological implications.

3. Lines 42-43: these three papers are not cited in the reference list

We fixed it, thank you.

4. Line 50: Or (2020) is not in the reference list

We fixed it, thank you.

5. Line 51: I don't understand how a modelling approach can be destructive? This should be clarified.

The modelling approach itself is not destructive. However, the determination of soil hydraulic parameters required to parameterize physically based models commonly relies on laboratory or field experiments involving soil sampling, excavation, or sensor installation, which can disturb the soil structure. This limits the feasibility of repeated measurements at the same location for long-term time-series analysis. We have revised the text to clarify this distinction.

Revised manuscript text (Lines 51–52):

"Additionally, soil hydraulic parameters for physically based models are usually obtained through soil sampling or sensor installation, both of which disturb the soil structure, limiting the feasibility of repeated measurements for long-term time-series analysis."

> 6. Line 77: I think you could also cite Robinson et al. (2019) here (*Global Change Biology*, 25, 1895-1904).

Thanks, we added the citation.

> 7. Lines 79-81: It may not be related to only soil structure. What about wettability (hydrophobicity)? Could that also play a role at these time scales? Perennial vegetation can also adapt to climate change. One example comes from an earlier modelling study based on the TERENO SoilCan data (see Jarvis et al. 2022. *Hydrology and Earth System Sciences*, 26, 2277-2299).

We agree that changes in soil hydraulic behaviour over time can result not only from structural evolution but also from changes in wettability, vegetation dynamics, and land-use transitions. Our statement was intended to emphasize that, given constant soil texture over decadal time scales, structural changes are expected to be a major contributor to shifts in the soil water response function. We have revised the manuscript text to acknowledge additional mechanisms explicitly.

Revised manuscript text (Lines 79–81):

"Considering that texture remains constant at time scales of decades, the changes in the response function are likely to be related to changes in soil structure and associated hydraulic properties driven by structural reorganization, wettability effects, root activity, or land-use changes rather than to textural change."

> 8. Lines 92-93: this statement is disproved by the following sentence (and also by the author's work). I think this sentence should be deleted (nothing important is lost).

We agree that the sentence is not essential and may cause confusion in light of the following statement. We have therefore deleted it from the manuscript.

> 9. Line 93: The authors of this paper are incorrectly specified in the reference list.

We corrected it, thank you.

> 10. Line 113: No need to start a new paragraph here.

We changed it.

> 11. Lines 250-252: this seems very subjective. Why choose these percentiles and these scores? How were they chosen, presumably by trial and error?

The thresholds defining wet, moderate, and dry water-content situations were not chosen arbitrarily but were selected through exploratory testing on the dataset analysed in this study. We evaluated several percentile combinations and found that the 30th and 70th percentiles provided a clear separation of water-content regimes while maintaining a sufficient number of observations in each class for stable model training and evaluation.

Using narrower or more extreme thresholds (for example the 25th/75th or 95th/99th percentiles) substantially reduces the number of data points in the extreme classes. This leads to sparsely populated categories and increases the risk of overfitting to rare events rather than capturing general system dynamics. For this reason, we chose broader regime classes and allow the model to resolve variability within wet and dry ranges through the continuous input features, rather than introducing additional discrete "extreme" categories.

We have added a brief clarification in the manuscript to explain that the chosen percentile thresholds are specific to the dataset analysed in this study and were selected to balance regime separation and data availability:

Line 250-253

"These thresholds were selected to provide clear regime separation while ensuring sufficient observations per class for robust model training."

Also, there is no numerical reasoning behind assigning "wet = 1" rather than "wet = 10". The number does not represent how wet the soil is in a physical sense. It simply labels which water-content situation the soil is in at a given time step.

In the current implementation, this label enters the neural network as a numerical input. We therefore acknowledge that the chosen numeric values can influence how the model internally weights this feature. However, no physical meaning, ordering, or quantitative distance between regimes is intended, regardless of the specific numeric spacing used. The values are arbitrary identifiers applied consistently to distinguish discrete water-content situations, and no intermediate values occur in the data.

12. Line 371: write "water contents" rather than "values" (if I understood correctly)

We have replaced "values" with "water contents values" to improve clarity and specificity.

13. Line 425: "differently" not "different"

Thank you, we changed it.

14. Line 426: this seems too speculative. You don't know that it is related to structure. Maybe you could write ".... which may indicate"

We agree that the original phrasing implied a stronger causal interpretation than warranted by the data. We have revised the sentence to adopt a more cautious formulation, as suggested.

Lines 423-426:

" This indicates that for the coarse-textured soil included in this study (i) the effect of changing climatic conditions was rather small (very good NSE classification for both sites), but (ii) that these coarse-textured topsoils do not show identical response to the extreme year, as each lysimeter reacts slightly differently, which may indicate small differences in hydraulic properties."

15. Line 444: "that" not "who"

Thank you, we corrected it.

16. Lines 582-586: This is interesting, but the authors neglect the fact that WRC's measured in the laboratory do show strong correlations with texture across large regions showing

strong climate contrasts. This empirical support for texture-based estimates of the WRC is really very strong, especially in the dry region of the WRC. How can you reconcile your results with this past experience and knowledge?

We agree that laboratory-measured soil water retention curves (SWRCs) exhibit strong correlations with soil texture across large climatic gradients, particularly when compared to hydraulic properties in the wet range, and our results do not contradict this empirical relationship. Instead, our findings indicate that under natural field conditions, the effective SWRC inferred from long-term in situ soil moisture dynamics can deviate from static, laboratory-based texture estimates due to structural evolution and climatic history. In this sense, laboratory SWRCs describe intrinsic textural control, while our analysis captures additional field-scale dynamics that are not represented in texture-based estimates. We have clarified this distinction in the manuscript. In addition, as already stated in the beginning of this reply letter, the effect of soil structures in the dry range may be more relevant than expected and may also depend on the measurement method (and the use of disturbed or undisturbed samples). As was shown in Hohenbrink et al. (2023 Earth Syst. Sci. Data, 15, 4417–4432) using HYPROP method to analyse the soil hydraulic properties of undisturbed samples, there was a huge variability of conductivity and retention value also in the dry range that could not be captured by texture only.

We will add a sentence in the introduction talking about the dry range also lines were edited as follows:

Lines 584-586:

"Such context-dependent behaviour highlights the limitation of the common assumption that soils with the same texture will show comparable retention across regions, an assumption often made in the absence of better descriptors. While laboratory-measured SWRCs show strong and well-established correlations with texture across climatic gradients, particularly in the dry range, experimental evidence collected under natural field conditions indicates that this simplified description does not always hold (Hannes et al., 2016; Robinson et al., 2016; Aqel et al., 2024)."

17. Line 600: this is much too speculative. You need to write this more carefully ("may have"). There are other more plausible interpretations. It could simply be that hydraulic conductivity at and close to saturation has increased. Near-surface water contents during rainfall events of any given intensity would then be smaller. It could also be a consequence of "by-pass" water flow in shrinkage cracks.

We agree with the reviewer that the mechanistic interpretation of the observed carry-over effect was speculative and not directly supported by the analyses presented in this study. As this interpretation is not essential to the main objectives of the manuscript, we have removed the sentence to avoid over-interpretation.

Revised manuscript text (Lines 597–602)

"However, a clear carry-over effect was observed: soil water in the upper 10 cm was not fully replenished during the wet phase of autumn and winter 2019 and only reached comparable, though slightly lower, values in winter 2020. A comparable multi-year legacy across the full soil column was reported in the TERENO-SOILCan lysimeter network by Groh et al. (2020)."

18. Line 636: what is meant by "functional integrity"?

the term "functional integrity" was too vague in this context. We have replaced it with a more explicit description referring directly to soil hydraulic behaviour.

Lines 631–638:

"The classification outcomes across all lysimeters highlight the role of site memory and structural resilience in maintaining hydraulic behaviour under climatic stress. Soils assessed at their origin were more frequently classified as 'stable' or 'resilient' (e.g., Selhausen at Selhausen), while those translocated to different locations were more likely to be classified as 'changed' (e.g., Sauerbach at Bad Lauchstädt). These patterns indicate that soil structure, once adapted to specific climate regimes, may exhibit altered hydraulic responses when exposed to new environmental conditions. The presented methods allow detection of emerging shifts in soil hydraulic behaviour that may be relevant for soil health assessment and could serve as indicators of deteriorating soil health status."

19. Lines 643-645: yes, especially if they are statistical models. Isn't it reasonable to expect that physics-based models might perform better outside the training data?

Although physics-based models are commonly regarded as better suited for extrapolation beyond calibration conditions due to their explicit process representation, this assumption does not generally hold when relevant stress states are absent from the calibration data. Under such circumstances, key process parameters remain weakly constrained, and model performance can deteriorate when applied to conditions involving water or heat stress, as shown by Groh et al. (2022 Vadose Zone Journal, 21, e20202). Our statement was intended to refer primarily to data-driven or statistically trained models. We have revised the text to clarify this distinction.

Revised manuscript text (Lines 642–648)

"As remote sensing missions increasingly provide continuous global soil water content estimates, the proposed framework could be adapted for large-scale assessment of soil system stability. Furthermore, under scenarios of future climate change, where shifts in precipitation patterns and evaporative demand are expected, data-driven models trained on historical data may become progressively outdated. The presented residual-based approach (quantifying MB) enables early detection of such divergence, offering a method for identifying when model retraining or reparameterization is needed to maintain predictive reliability under non-stationary conditions."

20. Lines 650: The extent to which the subsoil is exploited by roots should be much more important for plant water supply than the surface 10 cm.

We agree that soil moisture at 10 cm depth does not represent full root-zone water availability and therefore should not be interpreted as a direct measure of plant water supply or agricultural impact. We have revised the sentence to restrict the statement to near-surface hydraulic conditions and soil water dynamics, which are directly supported by the measurements in this study.

"Temporal variations in the water content of the topsoil control near-surface hydraulic conditions, infiltration, evaporation, and gas exchange, thereby shaping soil water dynamics and physical soil functioning. Reliable information on soil water content dynamics in response to atmospheric conditions is thus essential to detect and anticipate critical changes in soil hydraulic behaviour."

21. Minus signs (for matric potential) are missing on all the WRC plots

Matric potential is plotted as its magnitude (|h|), following a common convention in soil physics. We have clarified this by updating the axis labels in all SWRC plots to "Matric potential |h| [m]", so explicit minus signs are not required.

---

## Author Comment (AC2)

**REVIEW 2 (our reply in black)**

In this paper the authors use a neural network trained on meteorological variables, a seasonal trend analysis, and categorical definitions of soil conditions to quantify and evaluate soil water response across soil types and climate regions after an extremely hot summer in Germany.

Overall, I found the study and content of this paper compelling as well as the author's motivations for incorporating increasingly available remotely sensed products into soil stability analyses. However, I would like to see more discussion of the evaluation of the developed model and reasoning behind decisions, such as using Nash–Sutcliffe efficiency.

We thank the reviewer for this helpful comment. We clarify that in this study Nash–Sutcliffe efficiency (NSE) is not used as an optimization target or as a metric we aim to improve through model tuning. NSE was selected because it is a widely used, dimensionless performance metric in hydrological modelling that is sensitive to both magnitude and timing errors and therefore provides a convenient measure for comparing dynamic agreement across lysimeters.

In this study, NSE is used as a general indicator of how similar the soil-moisture dynamics at a tested lysimeter are to the reference soil–climate response function used for model training. Lower NSE values indicate reduced dynamic agreement but are not interpreted as diagnostic of a specific type or mechanism of change.

The detection and classification of changes in the soil water content response function are based on the temporal evolution of mean bias, which quantifies whether the tested system progressively drifts closer to or farther from the trained reference response over time. Accordingly, NSE characterizes overall dynamic agreement, while mean bias captures the direction and progression of change.

We have added this clarification to Section 2.5 of the revised manuscript (after line 365).

"In this study, NSE is used to characterize the overall level of dynamic agreement between predicted and observed soil water content, while the detection and classification of temporal changes in the soil water content response function are based exclusively on the year-by-year evolution of mean bias."

Further, as the authors identify, performance struggles at the site on which the model was not trained. This paper could be made far more robust through the reverse analysis—training the model in wetter conditions—to see how comparable the results are. Or training the model on both a wet- and dry-regime lysimeter simultaneously.

We explored reverse-regime training in preliminary analyses across multiple lysimeters to test whether the conclusions depend on the chosen training regime. In these tests, the model was trained under wetter conditions, and the overall conclusions remained unchanged. In the figure below, we present a reverse-regime comparison for the relocated soil shown in Figure 7 of the manuscript, representing a soil transferred from its original site to a wetter climatic regime. When the model is trained on a wet-regime reference lysimeter, the agreement between observed and simulated water content under wet climatic forcing improves, as shown by comparison of the time-series predictions in Figure R1 (a) with those in Figure 7(a) in the main manuscript. However, the temporal evolution of mean bias remains consistent with the original analysis, as evidenced

by the similar bias patterns in Figure R1(b) and Figure 7(b). As a result, the interpretation and classification of the site remain unchanged: the site continues to exhibit persistent drift relative to the reference response and is therefore classified as "changed". In other words, training on a wetter regime improves water-content prediction in a wet climate, but it does not alter the outcome of the change-detection framework.

We agree that training the model on one wet-regime and one dry-regime reference lysimeter would, in principle, further improve predictive performance across contrasting climatic conditions. However, the objective of this study is not to maximize prediction accuracy, but to detect deviations from a reference soil–climate response function. For this purpose, the training lysimeter must provide a clear and stable reference response. In practice, we assess this stability by examining the soil-water retention curve (SWRC), ensuring that the training site exhibits a well-defined relationship between matric potential and water content, and avoiding cases where a wide range of water content occurs at the same matric potential. Training on a lysimeter that contains multiple overlapping SWRC regimes would lead the model to learn a blended reference response, reducing the interpretability of deviations as change signals. To avoid capturing more internal variability than required for change detection, we therefore deliberately train the model on a single lysimeter exhibiting a stable and well-defined hydraulic response.

[Figure]

*Figure R1 Analysis of soil water content dynamics (2015–2023) for the relocated Bad Lauchstädt-origin soil lysimeter tested at Selhausen under reverse-regime training. Reverse-regime training denotes training the model on a wet-regime reference lysimeter instead of the dry-regime reference used in the main analysis. (a) Comparison of measured (blue) and simulated (orange) daily water content values when the model is trained on a wet-regime reference lysimeter, showing improved agreement under wet climatic conditions. (b) Mean Bias (MB) started slightly negative in 2015, increased towards positive values in 2019–2020, and subsequently exhibited interannual fluctuations in later years, displaying a temporal pattern comparable to the original analysis. (c) Soil-water retention curves (SWRCs) from the wet-regime training site (grey) and from the relocated Bad Lauchstädt lysimeter for selected years with low MB (2015, 2016) and high MB (2019, 2020) show that the inferred hydraulic response and site classification remain consistent with the original training approach.*

Finally, we note that interpreting persistent bias drift as evidence of changes in the soil water content response function assumes that climatic forcing is correctly represented in the model. Under the lysimeter conditions of this experiment, where precipitation and atmospheric demand are directly measured and the water balance is well constrained, such persistent deviations are

most plausibly linked to changes in soil hydraulic behaviour, although other unobserved factors may also contribute.

**Specific comments**

- In Line 75, I'd like to see additional citations to back up the claim of 'experimental studies', implying multiple, showing these shifts in soil water content dynamics following extreme events. Or rephrase.

Experimental evidence is listed in Robinson et al., 2019 (DOI: 10.1111/gcb.14626). We stated this explicitly in the revised manuscript: "Experimental studies have shown that extreme events such as drought can induce persistent shifts in soil water content dynamics, potentially leading to alternative stable states (see Robinson et al., 2016; Reinsch et al., 2024, Quintana et al., 2023 and the section on illustrative examples in Robinson et al., 2019,)."

- Clarifying question, beginning at the end of Line 79, are you stating that soil texture remains constant through land-use changes? Or in your modeling approach? Or something else?

In this statement, we do not refer to a modelling assumption, nor do we imply that soil texture changes through land-use transitions over the time scales considered. Instead, we refer to a well-established physical property of soils: the particle-size distribution (soil texture) that changes only over much larger time scales (at least decades to centuries for clay translocations and much longer for weathering), whereas soil structure and pore organization may evolve over years to decades due to biological activity, management, compaction, or climatic stress.

Therefore, in the context of this study, changes in the soil-water response function are interpreted as arising primarily from structural and hydraulic modifications rather than from changes in textural composition. We have revised the sentence to make this physical interpretation explicit.

"Considering that texture remains constant at time scales of decades, the changes in the response function are likely to be related to changes in soil structure and associated hydraulic properties driven by structural reorganization, wettability effects, root activity, or land-use changes rather than to textural change."

- Line 118: While the authors state that predictive modeling providing accurate predictions of soil water dynamics is "not the focus in this study," I do think that claim requires substantial citations. Alternatively, they could discuss their choice to use the NSE classification here as well as add more information about the pros and cons of the NSE method as this is missing and needed, when the strength of the paper relies on NSE performing as described. Additionally, this sentence (Line 115-119) is quite unwieldy, and it isn't clear what point the authors are trying to make.

We agree that the original sentence was unwieldy and did not clearly convey the role of predictive modelling in our study. We have rewritten this passage to clarify that, although the model produces predictions of soil water content, the primary objective is not prediction accuracy itself, but to use deviations between observed and simulated behaviour to detect changes in the soil–climate response function.

Specifically, we replaced Lines 115–119 with the following text:

"In this study, we adopt a predictive modelling approach in which deviations between observed and simulated soil water content over time are used to identify changes in the soil–climate

response function. While the model also produces predictions of soil water content, prediction accuracy is not the objective; instead, model–data deviations are used to detect temporal drift in soil water response behaviour. The overall level of dynamic agreement is summarized using Nash–Sutcliffe efficiency, while the detection and classification of changes rely on the temporal evolution of mean bias, as described in Section 2.5."

In addition, Section 2.5 now provides the definition of Nash–Sutcliffe efficiency and explains its role as a measure of dynamic agreement, as well as its complementary use with mean bias to identify the direction and progression of change. This revision clarifies the intended role of predictive modelling in the study and the rationale for using NSE in the change-detection framework.

- Figure 4: Considering the study focuses on the summer of 2018, would highlighting that data be evocative rather than the other groupings? It is not clear why the different time periods have been selected as they are not all the same length. Is this arbitrary or is it related to the mean bias results? Regardless, I think some minor explanation is needed. If the purpose of Figure 4 is only to motivate the selection of a particular lysemeter, the spread of values in the contrasting lysimeter is more than enough, without the additional colors).

The time periods shown in Figure 4 were not selected arbitrarily. They were chosen to reflect the hypothesis that the summer drought of 2018 acted as a trigger for changes in soil-water response dynamics. We do not expect such changes to occur instantaneously or to be confined to a single year; rather, if a shift in hydraulic behaviour is induced by an extreme event, the soil may require multiple seasons to adapt to a new dynamic regime or to recover toward its original state. The selected periods therefore represent pre-event, transition, and post-event phases, allowing visualization of how soil-water behaviour evolves before and after the 2018 drought.

We agree that highlighting the summer of 2018 is evocative given its central role in this study. However, the purpose of Figure 4 is to motivate the selection of a reference training lysimeter by illustrating differences in stability versus variability of soil-water response across time. For this reason, we chose groupings that reflect dynamic phases rather than fixed-length calendar windows.

- Line 252: This may be a naive comment, but is there a reasoning behind the numerical values chosen to represent each categorical condition? For example, why is 'wet' 1 rather than 10?

There is no numerical reasoning behind assigning "wet = 1" rather than "wet = 10". The number does not represent how wet the soil is in a physical sense. It simply labels which water-content situation the soil is in at a given time step.

In the current implementation, this label enters the neural network as a numerical input. We therefore acknowledge that the chosen numeric values can influence how the model internally weights this feature. However, no physical meaning, ordering, or quantitative distance between regimes is intended, regardless of the specific numeric spacing used. The values are arbitrary identifiers applied consistently to distinguish discrete water-content situations, and no intermediate values occur in the data.

- Relatedly, on what basis are you defining wet, moderate, and dry using the 30th and 70th percentiles? Why not 25th and 75th? Is this arbitrary? Further, have you considered looking at 'extremely' wet or dry categories? Considering you call 2018 an 'extreme' year throughout the manuscript, I'd be curious to know how incorporating the 95th or 99th percentiles for drying and wetting would be reflected.

The thresholds defining wet, moderate, and dry water-content situations were not chosen arbitrarily but were selected through exploratory testing on the dataset analysed in this study. We evaluated several percentile combinations and found that the 30th and 70th percentiles provided a clear separation of water-content regimes while maintaining a sufficient number of observations in each class for stable model training and evaluation.

Using narrower or more extreme thresholds (for example the 25th/75th or 95th/99th percentiles) substantially reduces the number of data points in the extreme classes. This leads to sparsely populated categories and increases the risk of overfitting to rare events rather than capturing general system dynamics. For this reason, we chose broader regime classes and allow the model to resolve variability within wet and dry ranges through the continuous input features, rather than introducing additional discrete "extreme" categories

We have added a brief clarification in the manuscript to explain that the chosen percentile thresholds are specific to the dataset analysed in this study and were selected to balance regime separation and data availability.

Line 250-251:

"These thresholds were selected to provide clear regime separation while ensuring sufficient observations per class for robust model training."

- In the mean bias aggregation, did you also look at a seasonal (wet v dry season) mean bias, or only annual? I would be curious if the response function is stable across seasons as well (to ensure there isn't variability between wet and dry that are cancelling each other out, for instance). This seems to be most evident in Figure 7, where the predicted water content does not capture both the driest and wettest water content periods. In later years the mean bias improves, but the prediction only appears to have a slightly better fit.

In the present study, mean bias was evaluated on an annual basis. During preliminary analyses, we examined separating bias into wet- and dry-season subsets to test whether seasonal variability could influence the interpretation of bias evolution.

As is stated by the reviewer, computing seasonal bias removes the risk that opposite errors in wet and dry periods cancel each other. However, in our dataset, seasonal subdivision reduced the number of observations per subset and led to unstable bias estimates. To address this, we performed an additional trial using three-year seasonal bias aggregation, which increased data density per subset. This three-year seasonal bias followed the same overall trend as the annual bias, confirming that the detected drift is not an artifact of yearly aggregation. Nevertheless, because the available time series spans only about ten years, three-year aggregation yields very few bias points and does not provide sufficient temporal resolution to identify when changes occur. We therefore retained annual mean bias as the primary evaluation metric, while explicitly considering the potential for wet–dry error compensation raised by the reviewer.

Regarding the reviewer's interpretation of Figure 7, we agree that early years show limitations in reproducing both the wettest and driest conditions (Figure 7a). However, the improvement in

mean bias in later years is not only a numerical artifact: it corresponds to a clearer reduction of systematic offset between observed and simulated water content across the time series (Figure 7a), which is summarized by the upward shift of mean bias toward zero in Figure 7b. In other words, Figure 7b reflects a real reduction in systematic mismatch visible in Figure 7a, even though event-scale discrepancies remain.

Finally, we note that for two out of the twenty-four samples, the evolution of annual bias does not perfectly follow the shift in the soil-water retention curve shape. This occurs in cases where the model exhibits opposite-sign errors in wet and dry periods (for example, underestimation in one regime and overestimation in the other), so that these errors partially cancel in the aggregated annual bias, as highlighted by the reviewer. These cases illustrate a limitation of using aggregated bias metrics alone and reinforce the need to interpret bias together with predicted and observed time-series behaviour and the corresponding SWRC evolution.

We have added this clarification to Section 2.5 of the revised manuscript (after line 337)

"Deviations between predicted and observed water content can differ in sign across moisture conditions and event types, such that opposite errors may partially cancel when averaged over a full year. Therefore, mean bias is interpreted as an indicator of long-term temporal drift in model–observation differences, while soil water retention curves are used here (see next section) to verify the physical consistency and direction of detected changes; in cases where retention measurements are not available, robustness of bias trends can alternatively be assessed by repeating the analysis using different temporal aggregation periods."

- Line 423: How do you conclude that this is a result for coarse soil rather than all/many/some compositions of topsoil? When you say 'this coarse soil' are you simply referring to that translocated lysimeter? Or making a statement about coarse soils in general? If the latter, I don't think this is adequately supported.

We refer here specifically to the coarse-textured soil included in our lysimeter dataset, not to coarse soils in general. The number of soil types examined in this study is not sufficient to support broad generalization across all coarse-textured soils. Our statement is therefore limited to the soils investigated in the TERENO–SOILCan lysimeter network. We attribute the observed behaviour to soil texture rather than site-specific topsoil composition because the same response pattern was observed for this soil when exposed to two contrasting climatic conditions. To avoid misunderstanding, we will add a sentence in the manuscript clarifying that these findings apply only to the study soils and are not intended as a general statement about coarse soils.

Lines 423:" This indicates that for the coarse-textured soil included in this study (i) the effect of changing climatic conditions was rather small (very good NSE classification for both sites), but (ii) that these coarse-textured topsoils do not show identical response to the extreme year, as each lysimeter reacts slightly differently, which may indicate small differences in hydraulic properties."

- I'd recommend reviewing the language in section 3.2 for clarity, when discussing transposed soil and 'changed' climate states. For example in line 447 the use of 'unchanged climate' is confusing. Perhaps consider consistent language such as 'original' climate, rather than (un)changed to avoid confusion.

We have revised the wording in Section 3.2 to use consistent terminology, replacing references to "(un)changed climate" with "original climatic conditions" and "new climatic

conditions" when discussing soil translocation. Where relevant, we further specify these as "wetter" or "drier" climatic conditions to improve clarity and avoid ambiguity. The example sentence suggested by the reviewer has been adopted accordingly.

The sentence was edited to be:

"In short, these examples show that the Bad Lauchstädt soil remained resilient under its original climatic conditions at Bad Lauchstädt but changed when exposed to **the wetter climatic** conditions at Selhausen."

- I would caution against describing the differences in climate between the sites as 'climatic shifts' (e.g., Line 518). To me this implies a step change in the climate at a given location, not transposing soil between two sites. The timescale of translocating soil is more abrupt than most/all climatic natural and/or anthropogenic climate change-forced shifts.

Thank you, we changed this. Our intention was to describe differences in climatic conditions between the two study sites rather than temporal climatic shifts at a single location. We have therefore replaced the term "climatic shifts" with "contrasting climatic conditions between sites" to avoid ambiguity.

- Were additional training methods explored for this analysis? (e.g., Is one lysometer enough to train this model? Does fit improve or plateau when trained on multiple lysimeters? Would training the model on lysimeters from each climate perform better than one lysimeter from a single climate?)

Additional training strategies were explored in preliminary analyses. In particular, we tested training the model on a wet-regime reference lysimeter, and an example of these results was already presented earlier (figure R1). As expected, predictive performance at wetter sites improved. We therefore agree that training on both a wet-regime and a dry-regime reference lysimeter would, in principle, improve performance across contrasting climatic conditions.

However, the objective of this study is not to maximize prediction accuracy, but to define a single, well-characterized soil water content response function that serves as a reference for detecting change in other lysimeters. Using multiple reference lysimeters would introduce multiple response functions into the training target and make the definition of a unique baseline response more difficult.

If two reference lysimeters were to be used, the wet-regime site would need to satisfy the same selection criteria as the dry-regime reference used here, namely a stable response function with minimal temporal change in the SWRC. In addition, the wet- and dry-regime reference sites should ideally occupy distinct volumetric water content ranges. In our dataset, the stable wet-regime lysimeters exhibit approximately the same volumetric water content range as the dry-regime reference lysimeter. Consequently, combining both in the training set would risk mixing response functions rather than defining a clear reference behaviour. For this reason, we retained a single stable reference lysimeter for model training.

- Line 600: what evidence for the physical soil response do you have? I don't see how this could have been determined from the model or information presented prior. Was the autumn and winter of 2019 comparable to conditions in prior years? Would we necessarily expect a rapid rebound to previous year's conditions?

We agree with the reviewer that the mechanistic interpretation of the observed carry-over effect as a structural change in pore connectivity and aggregation was not directly supported by the model results or by analyses presented earlier in the manuscript. This statement was based on qualitative inspection of the soil moisture time series rather than on independent evidence of physical soil alteration. As this interpretation is not essential to the main objectives of the study, we have removed the mechanistic explanation from the manuscript to avoid over-interpretation.

Revised manuscript text (Lines 597–602)

"However, a clear carry-over effect was observed: soil water in the upper 10 cm was not fully replenished during the wet phase of autumn and winter 2019 and only reached comparable, though slightly lower, values in winter 2020. A comparable multi-year legacy across the full soil column was reported in the TERENO-SOILCan lysimeter network by Groh et al. (2020)."

Line 650: I would not classify the top 10cm of soil moisture evaluated in this paper as defining water availability to plants. Most plants have root systems far deeper than 10cm. It seems a stretch to connect these results to agricultural impacts as surface vegetation was not varied or evaluated.

We agree that soil moisture at 10 cm depth does not represent full root-zone water availability and therefore should not be interpreted as a direct measure of plant water supply or agricultural impact. We have revised the sentence to restrict the statement to near-surface hydraulic conditions and soil water dynamics, which are directly supported by the measurements in this study.

"Temporal variations in the water content of the topsoil control near-surface hydraulic conditions, infiltration, evaporation, and gas exchange, thereby shaping soil water dynamics and physical soil functioning. Reliable information on soil water content dynamics in response to atmospheric conditions is thus essential to detect and anticipate critical changes in soil hydraulic behaviour."

- 673: 'extreme climatic events' seems overly broad, as you're only focusing on a single intense drought, you cannot draw conclusions against other extreme events (e.g. extreme precipitation).

We agree that the term "extreme climatic events" was too broad given that this study focuses on the impact of an intense drought event. We have therefore replaced this wording with "intense drought" to reflect the specific conditions investigated.

- 675: how are you defining 'past' here? Did you evaluate how unique the 2018 drought was in these locations/for these soils?

By "past climatic conditions" we refer to the climatic history under which soil structural properties developed at the original site before translocation. Our intention is to express that exposure to a broader range of climatic conditions may lead to soil structures that are more resilient to subsequent drought, and that this inherited behaviour persists after translocation to a new climatic regime. We have rephrased the sentence accordingly to clarify this meaning and avoid ambiguity.

We rephrase the sentence to be:

"We argue that soils which developed under a broader range of climatic conditions may possess soil hydraulic properties that enhance resilience to subsequent drought, and that this inherited behaviour persists after translocation to a new climatic regime".

Regarding the uniqueness of the 2018 drought, we cite Xoplaki et al. (2025) as following in our manuscript: "Total precipitation in central Europe was at the lowest percentiles relative to the 1976–2005 distribution; .... The summer of 2018 in Germany was characterised by the most extreme combination of high temperatures as one of the warmest years on record and as having the lowest precipitation since 1881"

**Technical corrections**

- Inconsistent spacing in em-dash usage. There should not be any spaces between words separated by em-dashes (e.g. use is correct on line 22, but not correct on lines 54, 55, 77, 78.

  Thanks, we changed it accordingly

- Line 75: possibly missing an 'a' or time scale should be plural.

  Thanks, we changed it accordingly

- Line 101-102: 'for a set of lysimeters' I'm not sure 'for' is the correct preposition, perhaps 'with a set of lysimeters'? Unless I'm misunderstanding the function.

  Thanks, we changed it accordingly

- Line 114: for clarity, I'd recommend adding commas around 'for example' to set off the clause.

  Thanks, we changed it accordingly

- Line 116: appears to be an extra space between 'which' and 'the'

  Thanks, we changed it accordingly

- Line 160: I would recommend offsetting the place names with commas (e.g., "including both, a wetter site, Selhausen, and a drier site, Bad Lauchstädt, ..."). Alternatively, you could use parenthesis.

  Thanks, we changed it accordingly

- Please align tense, particularly in the methods section, where both present and past tense alternate quite frequently.

  Thanks, we changed it accordingly as much as possible

- Line 215 is stating the same information as Line 130. I recognize the authors are re-introducing the reader to the input variables, but I would recommend restructuring as to not imply that water balance has not been discussed previously. My suggestion would be to either re-introduce all inputs in this section or, better yet, move this discussion up near Line 130. Moving the input variable discussion higher up would also reduce a bit of

confusion with the title of subsection 2.2.1 as you aren't discussing the STD until the second paragraph. In that same vein, the beginning of the 2.2.1 subsection implies that the input variables were selected after data cleaning, which does not seem to be the case (as the input variables were selected for cleaning/alignment).

Thanks, we changed it accordingly as much as possible

- Line 298: Should the Adam optimizer be cited?

We will add a citation for it, thanks

- Line 322, 670: I'm not sure the 2 hyphen is needed in Nash–Sutcliffe efficiency, nor should the E be capitalized. It may also be more clear to state: 'we use the Nash-Sutcliffe efficiency coefficient (NSE, see eq. 5) as a general descriptor...'

Thanks, we changed it accordingly as much as possible

- Line 444: use 'which' or 'that' rather than 'who'

Thanks, we changed it accordingly as much as possible

- Lines 519, 520: hyphens used instead of em-dashes.

Thanks, we changed it accordingly

- Line 615: missing a period.

Thanks, we changed it accordingly

- Lines 639, 641: hyphens used instead of em-dashes.

Thanks, we changed it accordingly

Line 669, 672: the preceding bullets end with a period, whereas these two do not.

Thanks, we changed it accordingly

- Line 676, 676: include only one does before the (i) and make require plural.

 Thanks, we changed it accordingly